# Patient reported postoperative pain with a smartphone application: A proof of concept

**Bram Thiel**[1,2]*, **Marc B. Godfried**[1], **Elise C. van Huizen**[3], **Bart C. Mooijer**[3], **Bouke A. de Boer**[4¤], **Rover A. A. M. van Mierlo**[5], **Johan van Os**[6], **Bart F. Geerts**[7], **Cor J. Kalkman**[8]

**1** Department Anesthesiology, OLVG Hospital, Amsterdam, The Netherlands, **2** University Medical Center Utrecht, Utrecht, The Netherlands, **3** Amsterdam University Medical Center, Amsterdam, The Netherlands, **4** Department of Information Technology, OLVG Hospital, Amsterdam, The Netherlands, **5** Logicapps Inc., Amsterdam, The Netherlands, **6** Department of Business intelligence, OLVG Hospital, Amsterdam, The Netherlands, **7** Department of Anesthesiology, Amsterdam University Medical Center, Amsterdam, The Netherlands, **8** Departments of Anesthesiology and Intensive Care, University Medical Center Utrecht, Utrecht, The Netherlands

¤ Current address: Le Blanc Advies, Apeldoorn, The Netherlands
* b.thiel@olvg.nl

**Data Availability Statement:** All relevant data are within the manuscript and its Supporting Information files.

## Abstract

Postoperative pain management and pain assessment are still lacking the perspective of the patient. We have developed and studied a prototype smartphone application for patients to self-record postoperative pain. The main objective was to collect patient and stakeholder critique of improvements on the usability in order to develop a definitive version. The secondary objective was to investigate if patient self-recording compared to nurse-led assessment is a suitable method for postoperative pain management. Fifty patients and a stakeholder group consisting of ten healthcare- and ICT professionals and two members of the patient council participated in this study.

### Main outcome

Thirty patients (60%) found it satisfying or very satisfying to communicate their pain with the app. Pain experienced after surgery was scored by patients as 'no': 3 (6%), 'little': 5 (10%), 'bearable': 25 (50%), 'considerable': 13 (26%) and 'severe': 1 (2%). Forty-five patients (90%) were positive about the ease of recording. Forty-five patients (90%) could correctly record their pain with the app. Thirty-eight patients (76%) agreed that in-app notifications to record pain were useful. Two patients (4%) were too ill to use the application. Based on usability feedback, we will redesign the pain intensity wheel and the in-app pain chart to improve clarity for patients to understand the course of their pain.

### Secondary outcomes

The median patient recorded pain app score 4.0 (range 0 to 10) and nurse recorded numerical rating scale (NRS) for pain NRS 4.0 (range 0 to 9) were not statistically different (p = 0.06). Forty-two percent from a total of 307 patient pain app scores were $\geq$ 5 (on a scale from 0 no pain at all to 10 worst imaginable pain). Of these, 83% were recorded as 'bearable' while only in 18% of the recordings patients asked for additional analgesia. The results suggest that self-recording the severity of postoperative pain by patients with a smartphone

**Funding:** SIDN fund, public benefit organization for Dutch internet domain registration (Url: www. SIDNfonds.nl), partially funded the development of the smartphone application. The funders had no role in the study design, data collection and analysis, decision to publish or preparation of the manuscript.

**Competing interests:** We have read the journal's policy and the authors of this manuscript have the following competing interests: Rover A.A.M. van Mierlo is employee at Logicapps the company that built the application commissioned by OLVG. OLVG and Logicapps have no commercial interest in the application. The application is available free of charge for patients. This does not alter our adherence to PLOS ONE policies on sharing data and materials.

application could be useful for postoperative pain management. The application was perceived as user-friendly and had high satisfaction rates from both patients and stakeholders. Further research is needed to validate the 11-point numeric and faces pain scale with the current gold standards visual analogue scale (VAS) and NRS for pain.

## Introduction

Postoperative pain assessment is predominantly performed by nursing staff. Patients are asked to rate the severity of their pain on a visual analogue scale (VAS) or a numerical rating scale (NRS). Both VAS and NRS are currently the gold standards for pain assessment and are often used to compare for severe postoperative pain between hospitals [1, 2]. However, it is conceivable that what we now consider as a valid outcome in reality ignores the patient's perspective [3]. For example, nurses sometimes may decide to alter the patients reported score, e.g., the recorded NRS may be a 'negotiated' result, or they 'average' the patient-reported pain score with their own assessment of the observed pain [4]. Another problem of current postoperative pain management is the outside pressure from patient safety and healthcare improvement programs mandated by governments to reduce the number of patients with pain scores higher than NRS 7 without having clarity on how patients value these scores [5]. Postoperative pain management based only on NRS rather than the patient's valuation might lead to analgesic over-administration due to strict adherence to a NRS threshold for administering analgesics established by protocol [6]. Lastly, the willingness from nurses to record pain scores is possibly low due to their workload and experienced high administrative burden [7, 8].

Self-assessment and recording of postoperative pain by patients might be the solution to overcome the aforementioned problems. The results of a study evaluating the effectiveness of using a self-reporting pain board in 50 oncology patients suggested that self reporting reduces under-assessment and provides a reliable and effective means of assessing pain [9]. In addition, hospitalized patients like to be in control and are willing to contribute to their treatment and the recording of symptoms (e.g. pain) in their electronic medical record (EMR) [9, 10]. Yet, most hospitals do not provide options or tools for patients to contribute to their EMR.

In the present study we have collected data of patient and stakeholder experiences on the usability of a newly developed smartphone application for self-recording postoperative pain by hospitalized patients. The main objective was to collect recommendations and improvements to adjust the smartphone application to its final version. Furthermore, we have collected patients' self-recorded pain scores with the application and pain scores recorded by nurses to assess the agreement between scores, and to determine if patient self-recording of postoperative pain with a smartphone application might be a suitable tool for postoperative pain management.

## Materials and methods

### Study design

A prospective mixed method cohort study was conducted in OLVG Hospital a Dutch general teaching hospital situated at two locations in Amsterdam, the Netherlands. The study was conducted according to the principles of Good Clinical Practice and to the Declaration of Helsinki [11]. The OLVG Hospital medical ethics committee considered this study as not being covered by the Medical Research Involving Human Subjects Act (WMO). The study was approved by

the local institutional board of OLVG Hospital. Written informed consent was obtained from all participating patients. The study is registered with a summary of the study protocol in de Dutch Trial Register (Nederlands Trial Register, NTR) with number NL6565 [12].

The prototype application was built by Logicapps, Amsterdam, the Netherlands. The application was developed to be accessible free of charge for patients using smartphones with an Android operating system. During the study, the application was securely connected with the data server of Logicapps according to the standards of privacy protection mandated by Dutch law. Furthermore, a Conformité Européenne certificate (CE certificate) to indicate conformity with health, safety and environmental protection standards for products within the European Economic Area (EEA) was not mandated during the conduct of this study [13].

## Participants and recruitment

Patients, aged 18 years or older, undergoing elective non-outpatient surgery and being in possession of a smartphone with Android operating system were regarded as suitable participants. Patients not able to read or understand Dutch were excluded from participation in this study. Patients were asked to participate in this study during their visit to the outpatient pre-anesthesia clinic. Participants received an information-letter providing details about the study. In addition, a researcher helped the patients with downloading the application to their smartphone and provided instructions on how to use the application and about the course of the study. A patient sample size of 50 patients, based on sample size calculation for qualitative studies, was estimated to be sufficient for collecting end-user feedback [14, 15]. Patients were recruited in consecutive order of hospital admission.

Furthermore, based on their interest and experience, we asked 4 anesthesiologists, 3 physician assistants, 2 medical students, 1 software engineer and 2 patients from de OLVG patient council, to be part of a stakeholder group to comment on the application. We estimated that the opinions of 12 stakeholders should provide sufficient information [16, 17].

## Study procedure

During preoperative assessment patients willing to participate received a verbal- and 'hands on' instruction on how to use the application. After downloading the application from Google Play™, patients were required to fill in the following information upon opening the application; name, date of birth and gender. Patients were then asked to allow that the application could send notifications to their mobile devices. The patients' identity was verified and the application was connected to the research database after authorization by a researcher. Furthermore patients received a 'pencil and paper' questionnaire for the evaluation of the application (S1 and S2 Files).

During admission, patients received standard care provided by the ward nurses and were treated for their postoperative pain according to the local standards based on the Guidelines for treatment of Postoperative Pain of the Dutch society of anesthesiologists [18]. Moreover, postoperative pain intensity was regularly verbally assessed on a scale from 0 (no pain) to 10 (worst imaginable pain) by ward nurses at least once every eight hours and thereafter recorded in the patient's electronic medical record (EMR). This method of pain assessment is common practice in most Dutch hospitals and commonly referred to as a 'numerical rating scale'. The use of the application during the course of the study had no implications for the treatment of postoperative pain. Furthermore, the application did not provide any in-app advice on treatment or analgesic medication to the patients or to the medical staff.

After surgery, patients were discharged from the postoperative anesthesia care unit (PACU) under the condition that their pain was 'bearable' with a NRS lower or equal to 4. Back on the

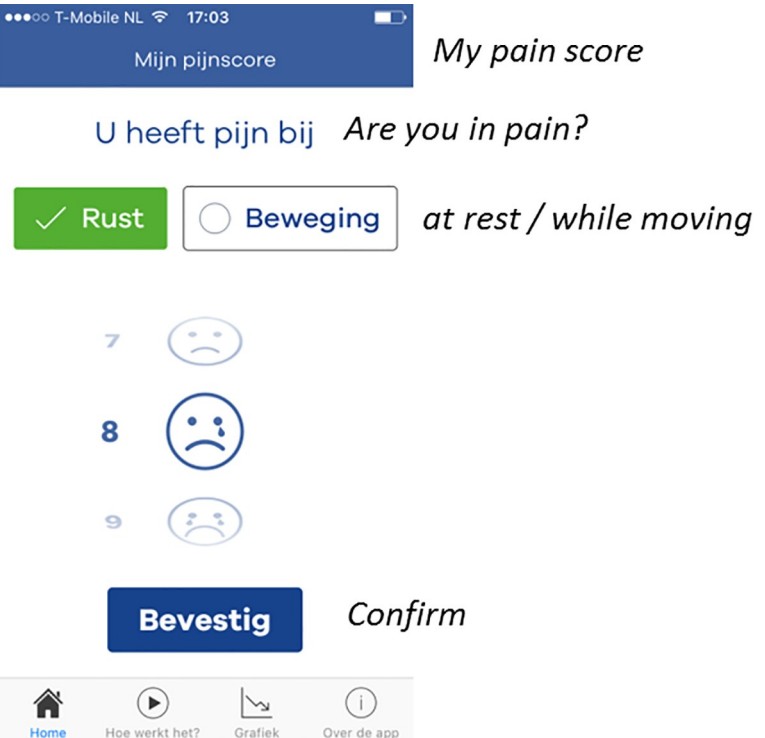

**Fig 1. Home screen: My pain score.**

nursing ward, patients could start using the application unrestricted to self-report on their postoperative pain. In addition, all patients were notified three times daily fixed at 8:00, 14:00 and 22:00 by the application to record pain. If the application was activated to record pain, the app opened on the home screen 'My Pain score' (Fig 1). In this screen patients could record their pain by answering the following question 'are you in pain: at rest or during movement?' Furthermore, they could define the intensity of their pain on a scale from 0 (no pain) to 10 (worst imaginable pain) by choosing the desired number with an on-screen numeric wheel. The default starting position of the wheel was set at 5 and the wheel was provided with text anchors at both ends of the scale referring to no pain (0) and worst imaginable pain (10). The pain intensity wheel was also equipped with a custom-designed 11-face scale to clarify and indicate the direction of the scale. The faces scale used in the application was designed to fit with the 11-point numerical scale and is based on the 6-faces scale recommended by the 2015 pain guideline from the Dutch college of General Practitioners [19]. The use of an 11-face pain scale seems appropriate for measuring acute postoperative pain in adult patients as was shown in a study amongst orthopedic surgical patients [20]. After recording their pain intensity, patients were asked to answer three additional questions (Fig 2A, 2B and 2C); 'Is the pain bearable?', 'Is extra pain relief required?' and 'Contact the nurse?' After these questions were completed, patients received a message from the application that their pain score had been recorded and that the in-app pain chart was updated accordingly (Fig 3). The first day after surgery, a researcher visited the patients on the ward to collect the questionnaire and had a short briefing of their participation.

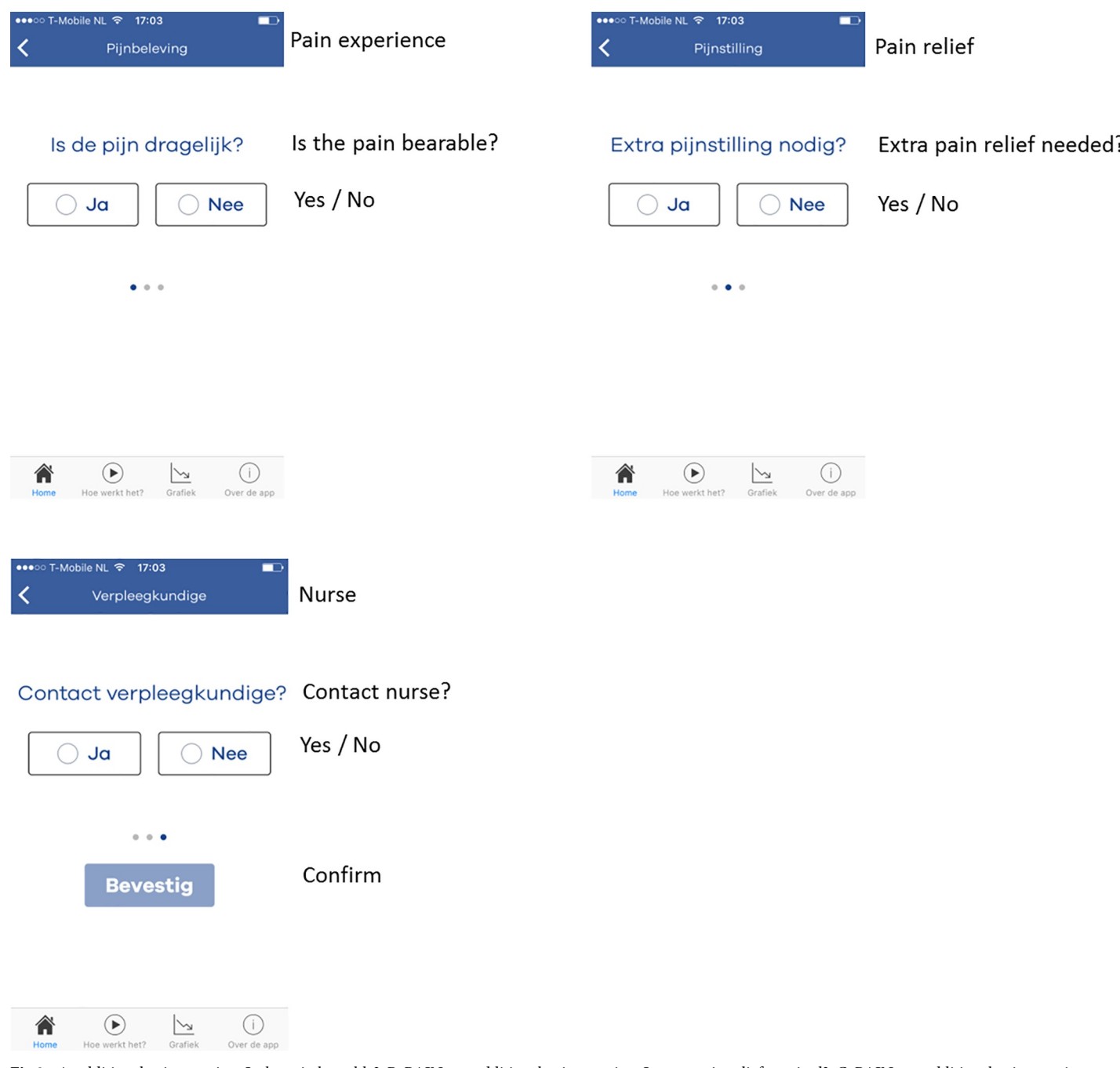

**Fig 2.** A. additional pain question: Is the pain bearable?. B. PAIN app additional pain question: Is extra pain relief required?. C. PAIN app additional pain question: Contact the nurse?.

## Outcomes

To compile an overview of patient characteristics and surgical procedures an information specialist extracted the following variables from the electronic medical record (EMR): age, type of surgery and hospital length of stay.

The main outcome was feedback from patients and stakeholders to improve the prototype application. We provided patients with a 12-item questionnaire (S1 and S2 Files). Questions

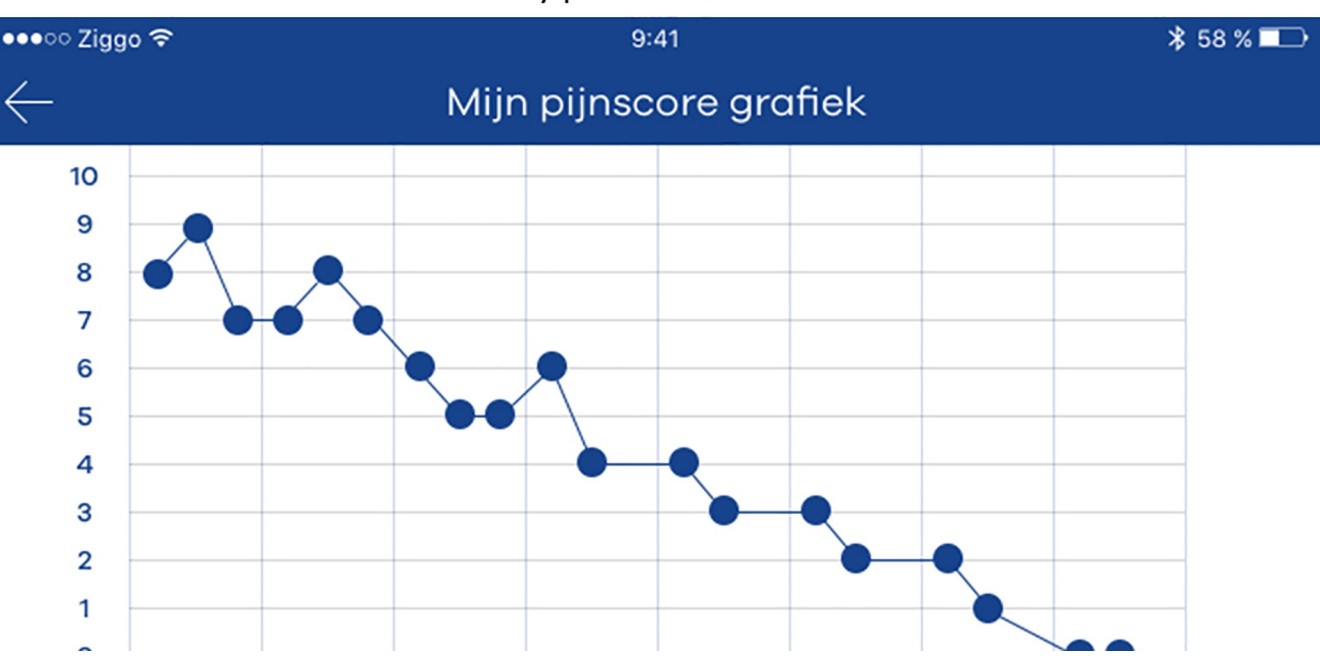

**Fig 3. In-app pain chart.**

had to be answered with a 5-point Likert-scale. In the questionnaire we have used the following categorization for pain; "no pain", "little pain", "bearable pain", "considerable pain", "severe pain" for patients to value their overall experienced pain during the admission [1, 21, 22]. Stakeholders were individually questioned during a semi structured interview conducted by a researcher after they had the opportunity to examine the application (S3 and S4 Files). Patient questionnaire and stakeholder interviews were derived from a previously conducted survey that evaluated quality assessment criteria and usability testing for smartphone applications intended for self-management of pain [23]. Questions were translated into Dutch language and assessed by our department of communication to check for comprehensibility and ambiguity.

The secondary outcome was to investigate differences in the number, intensity, and timing of the pain scores recorded by patients with the application compared to the pain scores recorded by nurses in EMR.

### Statistical analysis and reporting

Patient characteristics are described with categorical data presented in numbers and percentages and continuous data with mean and (interquartile) range depending on their distribution.

We grouped feedback from patients and stakeholders into the following themes: design, usability, content, and workflow, obtained from previous research [24–26]. In line with the methodology of framework analysis, 3 researchers (EvH, BT and MG) independently rated the comments and recommendations from the patients and stakeholders in order to create a ranking of necessary modifications ordered by importance and ease of adjustment [27].

Differences between patient reported pain scores and nurse reported pain scores were tested for significance using Mann Whitney U test. All statistical calculations were carried out

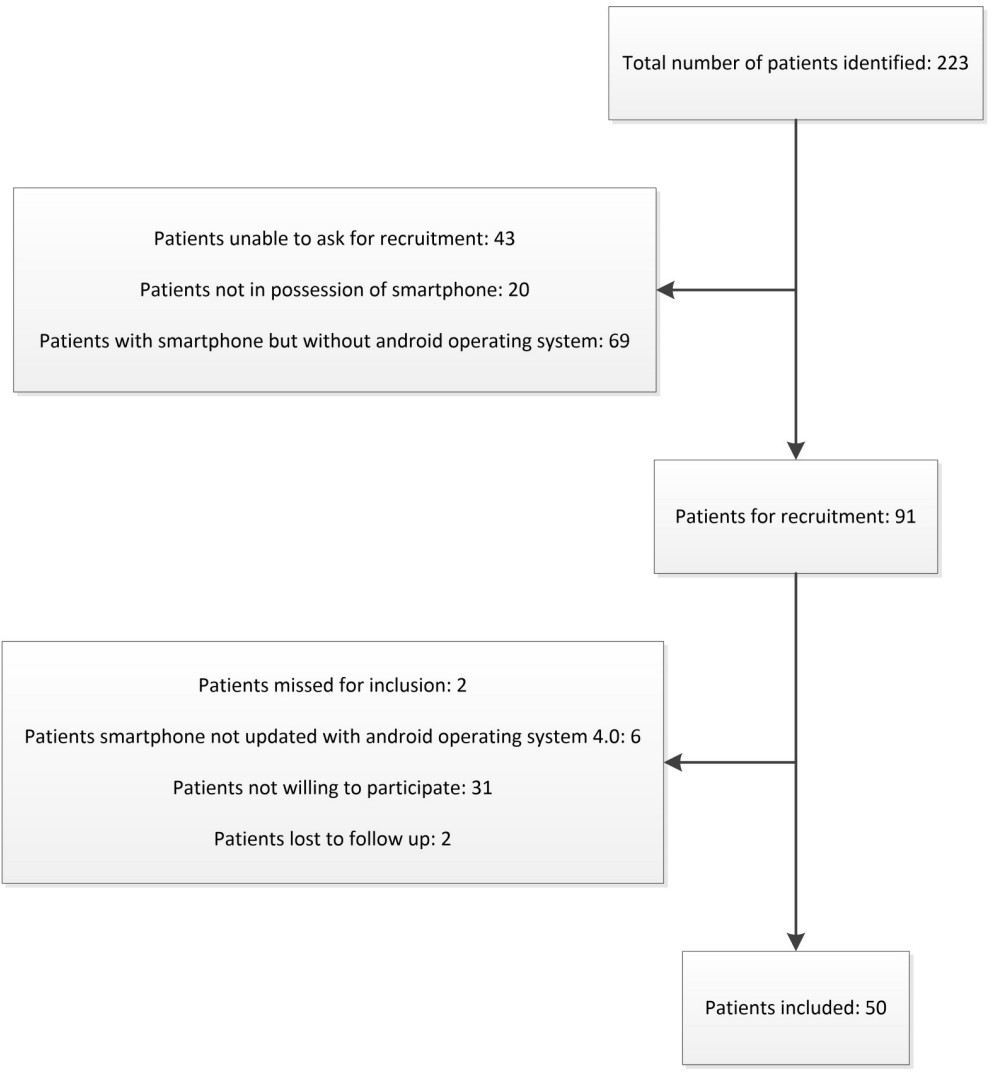

**Fig 4. Patient recruitment flowchart.**

using SPSS statistics version 20.0 (SPSS Inc., Chicago, IL). P-values less than 0.05 were considered statistically significant. The study results are reported according to the Strengthening the reporting of observational studies in epidemiology (STROBE) guideline and Standards for Reporting Qualitative Research (SRQR) [28, 29]

## Results and discussion

### Patient recruitment and demographics

The study was conducted between 3/10/2017 and 7/11/2017. We identified 223 patients who met the inclusion criteria (Fig 4).

A total of fifty patients possessing a smartphone with Android operating system 4.0 or higher were included. Two patients, who had signed informed consent, were not connected to the study database due to technical reasons. These patients were considered lost to follow up and were replaced.

**Table 1. Patient characteristics and surgical procedures.**

| | |
|---|---|
| **Total number of patients** | 50 |
| Age | 49 (21–72) |
| Male sex | 18 |
| Female sex | 32 |
| Postoperative hospitalization | 1.5 (0.3–8.9) |
| **Surgical procedures** | |
| Spinal surgery | 16 |
| Laparoscopic Gastric Bypass | 16 |
| Hip Arthroplasty | 5 |
| Knee Arthroplasty | 2 |
| Mamma Reduction | 2 |
| Tibia Osteosynthesis | 2 |
| Laparoscopic Hernia Repair | 2 |
| Parotidectomy | 1 |
| Tonsillectomy | 1 |
| Abdominoplasty | 1 |
| Septoplasty | 1 |
| Expansion Sphincter Pharyngoplasty | 1 |

Data are expressed in mean with range and in numbers

Two patients did not use the application and did not fill out the questionnaire. Stated reasons were they were feeling too ill after complicated laparoscopic gastric sleeve resection resulting in severe postoperative pain and severe nausea. One patient used the app but did not answer the questionnaire and could not be reached after hospital discharge. One patient used the application but only partially completed the questionnaire.

Eighteen patients participating were men and 32 were women. Mean age was 49 years (range 21 to 72 years). The severity of the surgery varied from minor (e.g. tonsillectomy) to major (e.g. hip arthroplasty). Postoperative hospitalization was 1.5 days (range 0.3 to 8.9) (Table 1). Indicating the experience patients have gained in using the app, the median number of pain app recordings before questionnaire completion was 3 (range 0 to 13).

## Main outcome

Thirty patients (60%) rated communicating the degree of pain with the application as satisfying or very satisfying (Table 2). The overall experienced postoperative pain was valued as no pain by 3 patients (6%), little pain in 5 patients (10%), 25 patients (50%) valued their pain as bearable and 13 (26%) valued their pain as considerable. One patient (2%) experienced severe pain. Asking patients if they could easily and correctly record their pain with the application 45 (90%) agreed or totally agreed. Asking about the three times daily notifications to score pain 38 patients (76%) agreed or totally agreed that this was useful. Regarding the overall appearance of the app 40 patients (80%) found it attractive or very attractive. Asking if it would be beneficial to contact a nurse with the application 9 (18%) of the patients reported that it would not be beneficial. The in-app pain intensity chart was valued as useful or very useful by 40 patients (80%).

All professionals agreed that the design of the application looked appealing and had neutral colors (Table 3). They shared the opinion that the in-app pain intensity chart was unclear. For example, there is no distinction between entered pain scores at rest or during movement and there is no information provided at what time point the pain was recorded.

**Table 2. Results patient questionnaires.**

| | | | | | |
|---|---|---|---|---|---|
| Did you like communicating the degree of pain with this app? | 6 (12%) very satisfying | 24 (48%) satisfying | 17 (34%) ok | 0 (0%) not satisfying | 0 (0%) very unsatisfying |
| How much pain did you experience after surgery? | 3 (6%) No | 5 (10%) little | 25 (50%) bearable | 13 (26%) considerable | 1 (2%) severe pain |
| The application is easy to use. | 28 (56%) totally agree | 17 (34%) agree | 1 (2%) no opinion | 0 (0%) disagree | 0 (0%) totally disagree |
| With the application I can correctly record my pain. | 21 (42%) totally agree | 24 (48%) agree | 1 (2%) no opinion | 0 (0%) disagree | 0 (0%) totally disagree |
| It is useful that the application reminds me to record my pain with notifications. | 19 (38%) totally agree | 19 (38%) agree | 6 (12%) ok | 1 (2%) disagree | 1 (2%) totally disagree |
| How do you rate the appearance of the application? | 7 (14%) very attractive | 33 (66%) attractive | 6 (12%) ok | 0 (0%) unattractive | 0 (0%) very unattractive |
| How do you rate the used colors? | 12 (24%) very attractive | 25 (50%) attractive | 9 (18%) ok | 0 (0%) unattractive | 0 (0%) very unattractive |
| How do you rate the used fonts? | 15 (30%) very attractive | 27 (54%) attractive | 4 (8%) ok | 0 (0%) unattractive | 0 (0%) very unattractive |
| How do you rate the layout of the application? | 17 (36%) very attractive | 23 (46%) attractive | 5 (10%) ok | 0 (0%) unattractive | 0 (0%) very unattractive |
| How do you rate recording your pain with the application? | 25 (50%) very useful | 16 (32%) useful | 4 (8%) ok | 0 (0%) not useful | 1 (2%) not useful at all |
| Would you like to be able to call a nurse with the application? | 16 (32%) very useful | 10 (20%) useful | 9 (18%) ok | 1 (2%) not useful | 9 (18%) not useful at all |
| How do you rate the in-application the pain chart? | 17 (34%) very useful | 23 (46%) useful | 5 (10%) ok | 1 (2%) not useful | 0 (0%) not useful at all |

Data are expressed in numbers and percentages; the difference in total numbers per question is explained by 3 patients who did not or partially complete the questionnaire

## Secondary outcome

Patients recorded 307 times their pain score with the app, while nurses recorded 396 times a NRS for pain. The median patient recorded pain app score was 4.0 (range 0 to 10), the median nurse recorded NRS for pain was 4.0 (range 0 to 9) (p = 0.06). The differences in pain recordings ranging from '0 to 4', '5 to 7' and '8 to 10' between patients and nurses were respectively 11% (p = 0.0024), 9% (p = 0.0096) and 2% (p = 0.27) (Table 4). One hundred ninety seven patient pain app scores (64%) were rated as pain during rest. Patients asked 92 times (29%) for extra analgesia and 63 times (20%) for a nurse. One hundred thirty one patients pain app scores (42%) were ≥ 5, of these, 109 were still rated as bearable and only 55 times they asked for extra analgesia (Table 5).

## Discussion

We have developed and tested a smartphone application for patients to self-record their postoperative pain as a first step of implementing patient self-recorded postoperative pain. Our main objective was to collect patient and stakeholder critique on the usability of the application in order to develop a definitive version of the application. Our secondary objective was to compare the self-recorded pain scores of the patients with the NRS pain scores recorded by nurses.

Both patients and stakeholders agreed that the application was easy to use and that its simplicity and design fitted the purpose of pain recording. Moreover, patients were willing and motivated to record their pain with the application. The difference in median pain intensity scores between those recorded by patients with the app and recorded by nurses were not

**Table 3. Stakeholder comments and recommendations for improvement.**

| Design | Recommendation | Rating |
|---|---|---|
| Modern look | No recommendation | |
| Various font styles | Standardize font style | + |
| Composed and clear | No recommendation | |
| Screens not properly aligned | Standardize alignment | ++ |
| Screen grouping and–proportion looks odd | Standardize grouping and proportion | + |
| Small navigation buttons at the bottom of the screens | Adjust buttons to bigger size | +++ |
| Buttons 'rest' and 'movement' are not the same size | Adjust buttons to equal size | +++ |
| Pain intensity wheel must be more prominent and bigger | Pain wheel in separate screen | +++ |
| Scale of the pain intensity wheel is not clear at a glance | Add information about the scale of the wheel | ++ |
| Too little distinction between the 11 faces scale | Use the frequently used 6 faces scale | ++ |
| Use of color and fonts is appropriate | No recommendation | |
| **Usability/Workflow** | **Recommendation** | **Rating** |
| Easy | No recommendation | |
| User-friendly | No recommendation | |
| Simple and quick | No recommendation | |
| It is unclear that the pain intensity must be confirmed with separate button | Add 'how to use' information | ++ |
| Progress authentication process is not clear | Add a waiting table | ++ |
| Not clear how many screens the app contains | Add screen bullets | +++ |
| Going backward in the app is difficult | Add 'how to use' information | +++ |
| It is possible to accidently skip screens and questions | Add 'how to use' information | ++ |
| **Content** | **Recommendation** | **Rating** |
| Pain intensity wheel is not clear | Highlight the chosen pain intensity | ++ |
| Clear and relevant questions | No recommendation | |
| Pain assessment questions: 'rest' and 'movement' are separate questions | Reformulate: are you bothered by pain during: rest, coughing, movement, not at all? | +++ |
| Pain assessment question: term 'bearable pain' | Reformulate into acceptable pain | ++ |
| Pain assessment question: Do you need extra analgesia? | Reformulate: Should something be done about your pain? | +++ |
| Pain assessment question: Contact nurse? | Reformulate: Should the nurse visit you? | +++ |
| Patient have to go through all questions even when not in pain | Start assessment with: Are you in pain? If not no other questions are asked. | +++ |
| If the patient indicates that the pain is not bearable | Automatically notify a nurse | ++ |
| Pain chart: no differentiation between pain at rest and during movement | Add separate lines in the chart for different recordings | +++ |
| Pain chart: not informative enough | Add date and time of the pain recordings and medication | +++ |
| Pain chart: axis is too small | Add a day chart and a week chart | +++ |
| In-app feedback: not personalized | e.g. add patient name | +++ |
| In-app feedback: not informative | Tailor feedback to pain recording, medication, add normal references of other patients | +++ |
| In- app feedback: not clear what actions are to be expected | Add information: what actions can a patient expect, e.g. a nurse is notified. | +++ |
| In-app feedback: disappears too quick | Adjust feedback time | +++ |
| **Additional comments** | **Stakeholder** | |
| "Is the nurse or acute pain service informed when the patient records severe pain in his EPD? This might be a good idea." | Anesthetist | |

(*Continued*)

**Table 3.** (Continued)

| Design | Recommendation | Rating |
|---|---|---|
| "Aren't you afraid that patients will manipulate or exaggerate their pain intensity scoring for quicker or better care?" | Pain nurse / Patient council member | |
| "It is important for nurses to recognize unbearable pain, but what to do when no extra analgesia is needed?" | Pain nurse | |
| "The usefulness of the Pain Chart for patients is questionable, too much focus on pain during admission." | Anesthetist | |
| "Add mean scores of other patients to the pain chart." | Patient council member | |
| "Adding mean scores from other patients could provide information for the patient about the course of pain 'am I doing well or not'" | Patient council member | |
| "Adding mean scores from other patients to the pain chart is not appropriate and could cause a negative effect on the course of pain if the patients score is under the mean." | Student software engineering / Anesthetist | |
| "It is questionable what the added value is of an in-app pain chart for patients, they might become too focused on their pain. Adding mean scores from other patients is a bad idea because: focus on their pain, pain is a unique and individual experience, placebo but also nocebo effects." | Anesthetist | |
| "The app must be available in more languages." | Patient council member / Anesthetist | |

The items mentioned in the table were independently rated by three researchers to determine the importance and ease of customization: + indicates low importance and/or difficult to customize; ++ indicates moderate importance and/or not very difficult to customize; +++ highly important and/or easy to customize.

statistically significant, therefore self-registration of postoperative pain with a smartphone application seems a comparable method to nurse pain assessment.

Our findings are in accordance with the results of several studies that have shown that pain applications are well used and appreciated by patients [24–26]. However, in the present study two patients stated that they were too ill to use the application due to severe postoperative pain and severe nausea. This is an important finding because it might influence the postoperative pain outcome if only patients with minor to moderate pain who are not very ill are willing to use the application. It also emphasizes that even with upcoming eHealth developments the ward nurse still has a very important role in our patient care. This is confirmed by the statement of some of patients in this study who denied when they were asked if it would be beneficial to call for a nurse with the app. One patient stated that 'When I need a nurse immediately I'll use the button next to my bed'.

The results of our study comparing overall pain intensity scores between patients and nurses might prove that self-recording is reliable method for pain assessment. However, there are some differences, the percentage of pain scores ranging from '0 to 4' recorded by patients with the app was lower compared with the NRS recordings of the nurses (p = 0.0024). Moreover, the percentage of pain scores ranging from '5 to 7' recorded by patients with the app was higher compared with the NRS recordings of nurses (p = 0.0096) although these differences are probably not clinically relevant.

Although our findings are promising, patients and stakeholders suggested several important improvements to the application. One, to redesign the pain intensity wheel by giving it a color or an arrow indicating the severity of pain. Both patients and stakeholders reported that they experienced difficulties in selecting a pain score. It was not always clear whether they

**Table 4. Differences in pain intensity recordings between patients and nurses.**

| Pain intensity | Patient recordings (307) | Nurse recordings (396) | Δ, Δ % (95%CI) | p (two-tailed) |
|---|---|---|---|---|
| 0–4 | 176 (57%) | 271 (68%) | 95, 11% (3.8 to 18) | 0.0024 |
| 5–7 | 105 (34%) | 100 (25%) | 5, 9% (2 to 16) | 0.0096 |
| 8–10 | 26 (8%) | 25 (6%) | 1, 2% (-2 to 7) | 0.27 |

Data are expressed in numbers and percentages. P value of 0.05 is considered statistically significant.

should rate their pain at rest or during movement. More importantly, the pain intensity wheel has not yet been validated in relation to VAS or NRS, which are currently considered the gold standards for pain assessment. Two, to add notifications to record pain as a red dot on the corner of the app. One patient recorded his pain only once, despite the fact that he was in the hospital for 5 days. The stated reason was that he had forgotten to record it. He said: 'I only watch the red notifications on the corners of the app icon. The notifications appeared as text messages and not as a red dot'. One of the professionals suggested adding a notification an hour after scoring a high pain score valued as unbearable or after the administration of medication. Three, to add more information to the in-app pain chart, such as pain at rest and during movement and a distinct time point of each pain score. The day to day pain intensity chart was added to provide patients their overview of the registered pain intensity scores. Most of the comments on the chart came from the interviews with the stakeholders they suggested that the chart might put too much focus on a patient's pain. This might result in aggravating behavior, when the pain does not decrease over time. Yet, the stakeholders suggested that it would be useful displaying analgesic medication in the application pain chart to show the patients the relation between medication (intervention) and their pain score. Four, to add more and clear personalized feedback messages in the app. Last, reformulate some of the in-app pain questions. For example: Are you in pain at rest or during movement? It might be better to start the pain assessment with the questions; 'Are you in pain?' followed by; 'How much pain do you have?' Then ask the patient if their pain is bearable and if they are bothered by their pain at rest, during breathing, during coughing, during movement or not bothered at all.

This study is not without limitations. First, the study was performed at two locations of OLVG Hospital in Amsterdam, which may not benefit the generalizability of the results. Second, only patients between 21 and 72 years old were included. However, the results of a study among 47 children and adolescents ranging from 9 to 18 years with cancer pain showed a high compliance rate and satisfaction ratings during a clinical feasibility test of a pain assessment smartphone application [26]. In addition, national survey's on the public adherence of mobile phones indicate that even the majority of elderly are in possession of a smartphone [30, 31]. That, unfortunately, does not prove that these elderly are able to use it during their hospital stay. In addition, patients who are too ill or in severe acute pain will be reluctant to use the application as we saw from our own data. Postoperative pain assessment performed by a nurse

**Table 5. Results in-app pain assessment questions.**

| Patient pain intensity | Bearable (yes) | Extra analgesia (yes) | Need for nurse (yes) | Pain at Rest |
|---|---|---|---|---|
| 0–4 (176) | 165 (93%) | 37 (21%) | 23 (13%) | 121 (69%) |
| 5–7 (105) | 95 (90%) | 38 (36%) | 30 (29%) | 63 (60%) |
| 8–10 (26) | 14 (54%) | 17 (65%) | 10 (39%) | 13 (50%) |

Data are expressed in numbers and percentages.

at regular intervals will therefore be a prerequisite. Third, we also acknowledge that the present study only included patients with a relatively short hospital stay.

The validity of the application in non-surgical patient populations still needs to be determined. This study is a first step in the process of developing an evidence-based smartphone application for pain recording. Real-time patient data, recorded with a smartphone, seems a promising method in better understanding the course of pain and pain management [32]. Currently there is an increase in the number of medical smartphone applications brought to the market without proper testing and scientific evaluation [23, 33]. Only few medical smartphone applications are designed with a value sensitive approach. As shown in other medical fields, in which new technology is being developed, value sensitive design seems to provide a workable basis to facilitate patient and healthcare profession values and integrate them into the final design [34]. Furthermore, it is still unclear what changes we have to expect in postoperative pain outcomes and how daily routine of nurses and doctors will change under the implementation of patients self-recording pain on a larger scale.

The process of designing and testing of the application is still ongoing. Version 2.0 of the application is currently in development in accordance with patients' and stakeholders' comments obtained with this study. Much work still needs to be done to thoroughly adjust and examine the psychometric features of the app's 11-point numeric and faces pain scale and validate it against VAS and NRS. Furthermore, we suggest that it is important to use the application to 'close the loop' in pain assessment and treatment. In the new version of the application we added the question: 'Should something be done about your pain?' If the patient answers 'yes', a comment for the nurses to discuss the pain appears in the patient electronic medical record. Really closing the loop between patients and nurses so that the patient is able to contact or warn the nurse with the application does require a technical IT solution which to date has not yet been realized.

In addition, we established a consortium of Dutch hospitals to study the possibilities of patient reported postoperative outcomes with a smartphone application for postoperative pain management and to be able to compare the anesthesia practices in different hospitals nationwide.

## Conclusions

The results of our study suggest that a smartphone application for self-recording postoperative pain by hospitalized patients is a user-friendly method with a high satisfaction rate for the majority of patients and stakeholders and that it could provide an outcome comparable to nurses pain assessment.

## Supporting information

**S1 File. Patient questionnaire, original.**
(DOCX)

**S2 File. Patient questionnaire, English translation.**
(DOCX)

**S3 File. Stakeholder interview, original.**
(DOCX)

**S4 File. Stakeholder interview, English translation.**
(DOCX)

**S1 Data.**
(XLSX)

**S2 Data.**
(XLSX)

## Author Contributions

**Conceptualization:** Bram Thiel, Marc B. Godfried, Bouke A. de Boer, Bart F. Geerts, Cor J. Kalkman.

**Data curation:** Bram Thiel, Elise C. van Huizen, Bart C. Mooijer, Johan van Os.

**Formal analysis:** Bram Thiel.

**Funding acquisition:** Bram Thiel.

**Investigation:** Bram Thiel, Marc B. Godfried, Elise C. van Huizen, Bart C. Mooijer.

**Methodology:** Bram Thiel, Marc B. Godfried, Bart F. Geerts, Cor J. Kalkman.

**Project administration:** Bram Thiel, Elise C. van Huizen.

**Software:** Rover A. A. M. van Mierlo.

**Supervision:** Marc B. Godfried, Bart F. Geerts, Cor J. Kalkman.

**Validation:** Bram Thiel.

**Writing – original draft:** Bram Thiel, Marc B. Godfried, Bart F. Geerts, Cor J. Kalkman.

**Writing – review & editing:** Bram Thiel, Marc B. Godfried, Bouke A. de Boer, Bart F. Geerts, Cor J. Kalkman.

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
