## [Decision Letter · Decision Letter 0]

22 Aug 2019

PONE-D-19-17106

Patient reported postoperative pain with a smartphone application: a proof of concept

PLOS ONE

Dear Thiel,

Thank you for submitting your manuscript to PLOS ONE. After careful consideration, we feel that it has merit but does not fully meet PLOS ONE’s publication criteria as it currently stands. Therefore, we invite you to submit a revised version of the manuscript that addresses the points raised during the review process.

We would appreciate receiving your revised manuscript by Oct 06 2019 11:59PM. To enhance the reproducibility of your results, we recommend that if applicable you deposit your laboratory protocols in protocols.io, where a protocol can be assigned its own identifier (DOI) such that it can be cited independently in the future. For instructions see: http://journals.plos.org/plosone/s/submission-guidelines#loc-laboratory-protocols

We look forward to receiving your revised manuscript.

Kind regards,

Erik Loeffen

Academic Editor

PLOS ONE

**Journal Requirements:**

2. Please include additional information regarding the survey or questionnaire used in the study and ensure that you have provided sufficient details that others could replicate the analyses. For instance, if you developed a questionnaire as part of this study and it is not under a copyright more restrictive than CC-BY, please include copies, in both the original language and English, as Supporting Information.

3. Please note that PLOS ONE has specific guidelines on software sharing (https://journals.plos.org/plosone/s/materials-and-software-sharing#loc-sharing-software) for manuscripts whose main purpose is the description of a new software or software package. In this case, new software must conform to the Open Source Definition (https://opensource.org/docs/osd) and be deposited in an open software archive. Please see https://journals.plos.org/plosone/s/materials-and-software-sharing#loc-depositing-software for more information on depositing your software.

"I have read the journal's policy and the authors of this manuscript have the following competing interests: Rover A.A.M. van Mierlo is employe at Logicapps Inc. the company that built the application commissioned by OLVG. OLVG and Logicapps have no commercial interest in the application. The application is available free of charge for patients. "

**Additional Editor Comments (if provided):**

Thank you for this revised version, incorporating the remarks I have made about conflict of interest (CoI). Although I appreciate you taking it seriously, in a next instant I would advise authors to just be transparant about the CoI's, not necessarily removing the author from the list (as all authors should have contributed in a way that justifies them being an author, removing someone who has done that amount of work is a bit harsh, and, if anything, might make the appearance around CoI more sketchy). It's not necessary to change this again now, but food for thought for a next time. For now, Van Mierlo is still listed as author in the manuscript itself, which should be removed in accordance with the article's metadata.

Regarding content, the manuscript does need quite some work to make it methodologically sound. Please see the reviewers' comments for guidance. I invite authors to send a revised version (with answers to reviewers). Good luck, and thank you.

Reviewers' comments:

Reviewer's Responses to Questions

**Comments to the Author**

1. Is the manuscript technically sound, and do the data support the conclusions?

Reviewer #1: Partly

Reviewer #2: No

2. Has the statistical analysis been performed appropriately and rigorously? 

Reviewer #1: Yes

Reviewer #2: No

3. Have the authors made all data underlying the findings in their manuscript fully available?

Reviewer #1: Yes

Reviewer #2: No

4. Is the manuscript presented in an intelligible fashion and written in standard English?

Reviewer #1: Yes

Reviewer #2: No

5. Review Comments to the Author

Reviewer #1: This is a study of a smart phone application for inpatient acute pain management. The authors show reasonable concordance with the phone app and nurses assessment.

In the future including the question, do you need additional pain medication could close the loop in assessment and treatments.

Major point:

As the authors note, …record their pain by answering the following question ‘are you in pain: at rest or during movement?’ Furthermore, they could define the severity of pain on a scale from 0 (no pain) to 10 (worst imaginable pain)….

There is typically a large difference between pain at rest and pain with activities. Depending upon the surgery, a VAS or NRS of 4 or more points. …”It was not always clear whether they should rate their pain at rest or during movement…” The same may apply to the nurses assessments.

This is a significant limitation of the study.

Other points:

Introduction: “the patient’s valuation possibly leads to analgesic over-administration and may even contribute to opioid addiction which is a major problem in health care.” We have not yet established short term inpatient opioid use causing opioid addiction. There are some associations but this is not yet established.

Reviewer #2: This manuscript describes a mixed methods study of the use and usability of a pain measurement application for cellular phones with Android operating systems. The manuscript is unclear on numerous occasions throughout and the methods have several serious flaws. The introduction needs to clearly identify the unique feature/s of the app or the study’s methods to support the importance of the work. The introduction needs to clearly identify the weaknesses of common post-operative pain measures and measurement methods that are likely to be specifically addressed by the app. What reasons for not participating were given by the 31 patients who declined to enroll? The methods describe that patients were not permitted to use their cell phones until their self-reported pain ratings were 4 or lower on a 0-10 pain scale. This methodology restricted the range of patients’ ratings using the app. The results need to provide descriptive statistics – including minimum and maximum - for the number of ratings provided per patient before questionnaire completion. Some patients did not stay for an entire day in the hospital so it is possible they had very little experience using the app before completing the survey about the app. Patients’ and nurses’ pain ratings were provided asynchronously so it does not make sense to test their agreement. In addition, why did the researchers choose to use nurses’ documented pain ratings as comparators to patients’ app ratings after they described how nurses alter patients’ self-reported pain ratings in the introduction? Regardless, nurse and patient agreement would be best compared by Bland Altman analyses with two groups of patients – one with the app and one with investigator recorded verbal pain ratings. The app uses an 11-level faces pain scale. The manuscript provides no reference for the reliability and validity of this faces pain scale. The methods provide little detail on the procedure for interviewing the healthcare professionals. For example, were they interviewed individually or in groups? It appears they were asked to examine the app at the start of the interview. Was this examination meant to be comparable to patients’ experiences actually using the app? Rationales for the results focus on the times of pain ratings and division of results by patient cohorts were not provided.

Minor Concerns

• Manuscript needs to clarify the terms “pain intensity slider” and “pain chart” early

• Abstract needs to clarify how patients’ and nurses’ pain ratings were “comparable”

• The terms “benchmark” and “benchmarking” appear throughout the manuscript without clear meaning

• The meanings of “back-office system of Logicapps,” “ease of customization,” “CE certificate,” and “validation questions” need to be provided

• Correct “a CE certificate for was not necessary…”

• The purpose of the references provided after the sentences describing sample size are unclear

• Throughout the manuscript, pain ratings are categorized (e.g., “considerable”) without any reference supporting the categorizations

• Are “bearable” and “acceptable” meant to be the same category of pain ratings?

• Are the “stakeholders” and “experts” also the “healthcare professionals”? Manuscript states “patients and stakeholders” and “patients and experts.” Participants, who were patient advisors, should not be described as “healthcare professionals” unless they were also licensed healthcare professionals.

• Correct “from the patients ands experts,” “chart does not,” “patients (green) record”

• The figures show the app asked if patients wanted additional medication, but they did show the app asking if the patients wanted nursing assistance as was stated in the manuscript

• Need to adhere to a guideline for reporting qualitative methods such as O’Brien et al. (2014) in addition to the STROBE guideline

• Manuscript refers to the reliability of the app, but the reliability of the app was not examined in the manuscript

• Correct “Although the healthcare professionals suggested displaying analgesic medication, in the in-app pain intensity chart as well”

• All the figures need to be in English

6. PLOS authors have the option to publish the peer review history of their article (what does this mean?). If published, this will include your full peer review and any attached files.

Reviewer #1: No

Reviewer #2: No

---

## [Author Response · Author response to Decision Letter 0]

6 Oct 2019

Rebuttal letter: PONE-D-17106

Title: Patient reported postoperative pain with a smartphone application: a proof of concept

Date: October 6th 2019

Dear Editorial Board, Dear Erik Loeffen,

On behalf of all authors who contributed to this manuscript I thank you for the opportunity to submit a revised version. We thank the editor and both reviewers for their valuable remarks and comments. In this rebuttal letter we will address all comments separately. We would also like to mention that the tables and reference list are only correct in the clean version copy, as we made changes to it which could not adequately be processed in the track changes version. 

Editorial comments: 

 To enhance the reproducibility of your results, we recommend that if applicable you deposit your laboratory protocols in protocols.io. 

A summary of the study protocol is published in the Dutch trail register www.trialregister.nl. The Nederlands Trial Register (NTR) is a publicly accessible and freely searchable register in which current prospective studies run in the Netherlands or that are carried out by Dutch researchers. The study registration number is: NL6565. We added the following sentences to the manuscript: “The study is registered with a summary of the study protocol in de Dutch Trial Register (Nederlands Trial Register) with number NL6565”, accompanied with the NTR website URL in the reference list. We think this provides enough information for researchers who are interested in the protocol and the study method. 

When submitting your revision, we need you to address these additional requirements (point 1).

We corrected and renamed the files for figures, tables and supplemental materials.

Please include additional information regarding the survey or questionnaire used in the study and ensure that you have provided sufficient details that others could replicate the analyses. For instance, if you developed a questionnaire as part of this study and it is not under a copyright more restrictive than CC-BY, please include copies, in both the original language and English, as Supporting Information (point 2).

The survey is not under a copyright. We derived questions and answer options from the study of Reynoldson et al 2014 and translated them into Dutch 1. This has already been mentioned in the original manuscript (page 8). The survey is uploaded as a Supporting Information file in Dutch (original language) and English. 

Please note that PLOS ONE has specific guidelines on software sharing for manuscripts whose main purpose is the description of a new software or software package. In this case, new software must conform to the Open Source Definition and be deposited in an open software archive (point 3). 

We understand the importance of sharing software. The objective of our study is to collect patient and stakeholder recommendations to improve the prototype pain application and to evaluate if patient self-reporting with an application is a workable method for postoperative pain management. We have no intentions to study and specify the technical background of the application. The application is tailored for OLVG hospital. The ‘source code’ of the application cannot be made publicly because of security and the risk of abuse by third parties.

Thank you for stating the following in the Competing Interests section: "I have read the journal's policy and the authors of this manuscript have the following competing interests: Rover A.A.M. van Mierlo is employee at Logicapps Inc. the company that built the application commissioned by OLVG. OLVG and Logicapps have no commercial interest in the application. The application is available free of charge for patients. "

Please include your updated Competing Interests statement in your cover letter; we will change the online submission form on your behalf (point 4). 

We confirm the following statement ‘This does not alter our adherence to PLOS ONE policies on sharing data and materials’ and added it to the new cover letter. We are very grateful if you change the online submission form on our behalf as you suggested. Furthermore, it was never our intention not to be transparent about the CoI. Removing the author seemed the best option to make the submission possible and was done at the insistence of Van Mierlo himself. We kept Van Mierlo listed as author without omitting the clarity of his competing interests. 

In your Data Availability statement, you have not specified where the minimal data set underlying the results described in your manuscript can be found. PLOS defines a study's minimal data set as the underlying data used to reach the conclusions drawn in the manuscript and any additional data required to replicate the reported study findings in their entirety. All PLOS journals require that the minimal data set be made fully available (point 5). 

We uploaded the study’s minimal underlying data set as a Supporting Information file.

Reviewer comments:

Reviewer #1: In the future including the question, do you need additional pain medication? Could close the loop in assessment and treatments. 

We agree with reviewer #1 that it is important to close the loop in pain assessments and treatments. In the version of the application that we are currently working on we added a similar question: ‘Should something be done about your pain?’ The formulation of this question is based on semantic arguments. Using the terms ‘additional pain medication’ could give the expectation to patients that pain medication always is an option and available. In reality, there are also other interventions such as, psychological support, posture and exercise advice, and applying heat and comfort. Closing the loop between patients and nurses does require in a technical solution so that the patient can contact or warn the nurse with the application this does require a technical solution which is not yet provided. 

Reviewer #1: As the authors note…record their pain by answering the following question ‘are you in pain; at rest or during movement? Furthermore, they could define the severity of pain on a scale from 0 (no pain) to 10 (worst imaginable pain)…There is typically a large difference between pain at rest and pain with activities. Depending upon the surgery, a VAS or NRS of 4 or more points…´It was not always clear whether they should rate their pain at rest or during movement…´The same may apply to the nurses’ assessments. 

To reviewer #1 this appears to be a limitation of the study but we see it is as a result from the study which we possibly have not clarified enough. This study was performed to collect patient and stakeholder comments to improve the application. One of the results is that patients are confused about whether they rate their pain in rest or during movement. One of the improvements of the new 2.0 version of the app is that it starts with the question ‘are you in pain?’ This is mentioned on page 19 of the manuscript. 

Reviewer #1: ‘’the patient’s valuation possibly leads to analgesic over-administration and may even contribute to opioid addiction which is a major problem in health care.’’ We have not yet established short term inpatient opioid use causing addiction. There are some associations but this is not yet established. 

We agree with reviewer #1 on this point and have revised this section of the manuscript.

Reviewer#2: The introduction needs to clearly identify the weaknesses of common post-operative pain measures and measurement methods that are likely to be specifically addressed by the app.

We have revised the introduction on several points. 

Reviewer#2: What reasons for not participating were given by the 31 patients who declined to enrol? 

According to Dutch legislation (Medical Research act). Patients can refuse to participate in medical research without giving reasons. Furthermore, if a patient refuses to participate, it is not allowed to document stated reasons and publish these. We do very much see the relevance for this manuscript in elucidating any potential (selection) bias but we regret we cannot and may not share the relevant data if we could.

Reviewer#2: The methods describe that patients were not permitted to use their cell phones until their self-reported ratings were 4 or lower on a 0-10 pain scale. 

This is not correct, as already stated in the manuscript.’After surgery patients were discharged from the postoperative anesthesia care unit (PACU) under the condition that their pain was ‘acceptable’ with a numerical rating scale (NRS) lower or equal to 4. Back on the nursing ward, patients could start using the application unrestricted to self-report on their postoperative pain’.

Reviewer#2: The results need to provide descriptive statistics-including minimum and maximum – for the number of ratings provided per patient before questionnaire completion. Some patients did not stay for an entire day in the hospital so it is possible they had very little experience using the app before completing the survey about the app. 

We agree with reviewer#2 that this provides a better insight into how long the patient has actually used the application. Unfortunately we have not asked patients to provide the questionnaire with data and time. The patient questionnaires were collected by one of the researchers on the day of discharge. 

Reviewer#2: Patients’ and nurses’ pain ratings were provided asynchronously so it does not make sense to test their agreement. In addition, why did the researchers choose to use nurses’ documented pain ratings as comparators to patients’ app ratings after they described how nurses alter patients’ self-reported pain ratings in the introduction?

The statistical analysis for comparing the differences in patient and nurse pain recordings is done with non-parametric testing using Mann Whitney U test as stated in the manuscript. As reviewer#2 already concluded the pain ratings were provided asynchronously so we did not test for agreement. The Argument for using nurse pain recordings to compare with is that there is no other alternative; nurse pain recordings are still the gold standard for post-operative pain assessment 

Reviewer#2: regardless, nurse and patient agreement would be best compared by Bland Altman analysis with two groups of patients – one with the app and one with investigator recorded verbal pain ratings. 

We are not sure if we understand this comment. Bland-Altman analysis is the quantification of the agreement between two quantitative methods by studying the mean difference and constructing limits of agreement under the condition that both methods are designed to measure the same parameter and that measurements are taken on the same time point 2. Separate patient groups as suggested by reviewer#2 is in our opinion not possible. 

Reviewer#2: The app uses an 11-level faces pain scale. The manuscript provides no reference for the reliability and validity of this faces pain scale.

The facial expression is added to the numerical rating scale to clarify the meaning of the numbers and the direction of the scale from 0 (no pain at all) to 10 (worst pain imaginable). The two scales are thus provided jointly. As we often see when we only use numerical rating scale many patients use the scale in reverse to rate their pain, as if to rate achievements and experiences with a number on a scale from 0 to 10, with 10 as the best positive result. We have added the following to the manuscript ‘The NRS was also equipped with an 11-face pain scale to clarify- and indicate the direction of the NRS. The use of an 11-face pain scale seems appropriate for measuring acute postoperative pain in adult patients as was shown in a study amongst orthopaedic surgical patients’ Provided with a suitable reference.

Reviewer#2: The methods provide little detail on the procedure for interviewing the healthcare professionals. For example, were they interviewed individually or in groups? It appears they were asked to examine the app at the start of the interview. Was this examination meant to be comparable to patients’ experiences actually using the app? 

The stakeholders were interviewed individually. Indeed, they were asked to examine the application before the start of the interview. This was not meant to be comparable to patients’ experiences who could actually use the application for the time being admitted to hospital. We revised the methods section of the manuscript to provide more clarity.

Reviewer#2: Rationales for the result focus on the times of pain ratings and division of the results by patient were not provided. We agree with reviewer#2 that the manuscript provides no clarity about the rationales for the secondary objectives. The secondary objective was to investigate if patient self-recording compared to nurse-led assessment is a suitable method for postoperative pain management. We have adjusted the abstract, introduction and materials and methods to explain our rationales better. 

Reviewer#2: Manuscript needs to clarify the terms “pain intensity slider” and “pain chart” early.

We corrected ‘pain intensity slider’ into ‘Numerical Rating Scale’ throughout the manuscript to provide more clarity. The in-app pain chart was already explained in the methods section with an accompanying image. We have not adjusted this. 

Reviewer#2: Abstract needs to clarify how patients’ and nurses’ pain ratings were “comparable”. 

We adjusted the abstract with the following sentence: The results suggest that the overall median pain intensity scores from patients recorded with a smartphone application compared with pain scores recorded by nurses’ show no statistical significant difference and therefore could be used for postoperative pain management.

Reviewer#2: The terms “benchmark” and “benchmarking” appear throughout the manuscript without clear meaning.

We have corrected the terms “benchmark” and “benchmarking” throughout the manuscript with appropriate reformulation. 

Reviewer#2: The meanings of “back-office system of Logicapps,” “ease of customization,” “CE certificate,” and “validation questions” need to be provided. 

We corrected ‘back-office system’ into ‘data server’. We corrected validation questions into ‘in-app pain assessment questions’. We corrected “ease of customization” into “ease of adjustment”

We explained ‘CE certificate’ in the manuscript as Conformite European certificate (CE certificate) this is a certificate to indicate conformity with health, safety and environmental protection standards for products used and sold within the European Economic Area (EEA) such a certificate is not mandated during development and research of a smartphone application.

Reviewer#2: Correct “a CE certificate for was not necessary…” 

We added Conformite European certificate (CE certificate) and corrected the sentence.

We thank reviewer#2 for the comments: The purpose of the references provided after the sentences describing sample size are unclear. In this reference the theory behind the methods used to estimate the sample size are explained

Reviewer#2: Throughout the manuscript, pain ratings are categorized (e.g., “considerable”) without any reference supporting the categorizations In the patient questionnaire we have used the following categorization; “no pain” , “little pain”, “bearable pain”, “considerable pain”, “severe pain”. This was meant as a verbal rating scale for patients to value their overall experienced pain during the post surgical admission. The original Dutch terms used in the questionnaire are derived from the studies of Breivik et al 2008 3 and Jensen et al 2011 4. Currently there is no validated verbal rating scale for Dutch language available. We have stated this in the outcomes section of the revised manuscript.

Reviewer#2: Are “bearable” and “acceptable” meant to be the same category of pain ratings?

We exchanged ‘acceptable’ throughout the manuscript in ‘bearable’ in accordance with the request in the application and patient questionnaire.

Reviewer#2: Are the “stakeholders” and “experts” also the “healthcare professionals”? Manuscript states “patients and stakeholders” and “patients and experts.” Participants, who were patient advisors, should not be described as “healthcare professionals” unless they were also licensed healthcare professionals.

We corrected ‘healthcare professionals’ throughout the manuscript into ‘stakeholders’ and explained the composition of this group in the abstract and methods section. 

Reviewer#2: correct “from the patients and experts,” “chart does not,” “patients (green) record”. 

We corrected these sentences in the manuscript.

Reviewer#2: The figures show the app asked if patients wanted additional medication, but they did show the app asking if the patients wanted nursing assistance as was stated in the manuscript.

We are not sure if we understand this comment all screens of the application are uploaded as a figure. Possibly due to the Dutch language used in de figures there might be some misinterpretation, for the revised manuscript we translated all figures into English. 

Reviewer#2: Need to adhere to a guideline for reporting qualitative methods such as O’Brien et al. (2014) in addition to the STROBE guideline. 

We agree with reviewer#2 that the suggested reference provides a useful checklist for reporting. We used it as a final check before submitting the revised version of the manuscript, Furthermore, we added the following to the manuscript ‘the study results are reported according to the Strengthening the reporting of observational studies in epidemiology (STROBE) guideline for observational studies and Standards for reporting qualitative research (SRQR)’. O’Brien et al 2014 5 has been added to the manuscript reference list. 

Reviewer#2: Manuscript refers to the reliability of the app, but the reliability of the app was not examined in the manuscript. 

We agree with reviewer#2, we did not test for reliability. We have changed the terms reliable and reliability into ‘comparable’ in several sections of the manuscript 

Reviewer#2: correct “Although the healthcare professionals suggested displaying analgesic medication, in the in-app pain intensity chart as well” 

We corrected this sentence into: ‘Yet, the stakeholders suggested that it would be useful displaying analgesic medication in the application pain chart to provide patients more information about their treatment’.

Reviewer#2: All the figures need to be in English.

We translated all figures to English where this was not sufficiently done so.

References

1. Reynoldson C, Stones C, Allsop M, et al. Assessing the quality and usability of smartphone apps for pain self-management. Pain medicine (Malden, Mass). 2014;15(6):898-909.

2. Bland JM, Altman DG, Measuring agreement in method comparison studies, Statistical Methods in Medical Research 1999, 8(2), 135–160. https://doi.org/10.1177/096228029900800204

3. Breivik H, Borchgrevink PC, Allen SM, et al. Assessment of Pain, British Journal of Anaesthesia 2008; 101(1);17-24

4. Jensen Hjermstad M, Fayers PM, Haugen DF, et al. Studies comparing numeric rating scales, verbal rating scales and visual analogue scale for assessment of pain intensity in adults: a systematic literature review. Journal of Pain and Symptom Management 2011;41(6):1073-93

5. O'Brien BC1, Harris IB, Beckman TJ, Reed DA, Cook DA, Standards for reporting qualitative research: a synthesis of recommendations, Acad Med. 2014 Sep;89(9):1245-51. doi: 10.1097/ACM.0000000000000388.

Sincerely yours, also on behalf of my co-authors

Bram Thiel, MSc

OLVG Hospital 

Department of Anesthesiology

PO Box 95500; 1090 HM Amsterdam, the Netherlands

E: b.thiel@olvg.nl; 

T: +31614584978

---

## [Decision Letter · Decision Letter 1]

18 Dec 2019

PONE-D-19-17106R1

Patient reported postoperative pain with a smartphone application: a proof of concept

PLOS ONE

Dear Thiel,

Thank you for submitting your manuscript to PLOS ONE. After careful consideration, we feel that it has merit but does not fully meet PLOS ONE’s publication criteria as it currently stands. Therefore, we invite you to submit a revised version of the manuscript that addresses the points raised during the review process.

Since the previous two reviewers were quite critical of the original manuscript, but unable to review your revisions, I’ve invited an additional reviewer to examine the manuscript. This reviewer noted several remaining major issues with the manuscript and with the revisions made in response to the previous review. I fully agree with the sensible and important comments from this reviewer. Based on the review, I’ve decided that the manuscript still needs to be thoroughly improved (major revision) to be considered for publication.

Should you decide to submit a revision, please carefully address all the issues raised by the reviewer in your response letter and make the required changes in the manuscript.

In addition, please pay particular attention to and thoroughly check the accuracy of all statistics throughout the entire manuscript. The reviewer already pointed out several errors or inconsistencies between the abstract and the manuscript body. I also noted a potential error with the numbers in the abstract. For instance, in the abstract you described that “…Two patients (1%) were …”, whereas this percentage should probably be 4% considering the sample of 50 patients. Finally, make sure that the track changes and clean version match exactly as there appear to be several discrepancies between both in the current versions, which hinders a careful review process.

We would appreciate receiving your revised manuscript by Feb 01 2020 11:59PM. To enhance the reproducibility of your results, we recommend that if applicable you deposit your laboratory protocols in protocols.io, where a protocol can be assigned its own identifier (DOI) such that it can be cited independently in the future. For instructions see: http://journals.plos.org/plosone/s/submission-guidelines#loc-laboratory-protocols

We look forward to receiving your revised manuscript.

Kind regards,

Peter M ten Klooster, Ph.D.

Academic Editor

PLOS ONE

Reviewers' comments:

Reviewer's Responses to Questions

**Comments to the Author**

1. If the authors have adequately addressed your comments raised in a previous round of review and you feel that this manuscript is now acceptable for publication, you may indicate that here to bypass the “Comments to the Author” section, enter your conflict of interest statement in the “Confidential to Editor” section, and submit your "Accept" recommendation.

Reviewer #3: (No Response)

2. Is the manuscript technically sound, and do the data support the conclusions?

Reviewer #3: Partly

3. Has the statistical analysis been performed appropriately and rigorously? 

Reviewer #3: No

4. Have the authors made all data underlying the findings in their manuscript fully available?

Reviewer #3: Yes

5. Is the manuscript presented in an intelligible fashion and written in standard English?

Reviewer #3: No

6. Review Comments to the Author

Reviewer #3: Thank you for the opportunity to review this interesting paper on using a smartphone to assess pain post operatively. Unfortunately, even though it has been revised, I find a couple major problems and many minor issues with this manuscript, most of which were not mentioned by the previous reviewers.

1. The revisions to the manuscript have added the label of NRS or Numerical Rating Scale to describe the app pain scale used. This is just not accurate. The Numerical Rating Scale is not just using the numbers 0-10 to describe pain. It should include these numbers along a line, with the whole line visible at once. Often there are anchor words at either end. The app, as shown in Figure 1, does not do this. I assume a line with number is what is used by the nurse. If instead the nurse used some other form of the numbers or it was purely verbal, it would be good to add this to the manuscript. Your Kim et al 2012 citation from the Journal of Palliative Care clearly shows a line with numbers scale where you can see the whole line at once. Your first citation, for the NRS, Breivik et al. 2008 from BJA also shows a line with number for the NRS.

Please refrain from referring to the app scale as the NRS. Much work has been done to validate the proper NRS as a good pain scale. Is there any scientific evidence of this wheel scale (where you can only see a few numbers at once and the faces seem to be the focus) being consistent and valid as a pain scale? What was the starting position of the wheel? Was it 0, 5, or 10? This needs to be well explained or cited within the paper.

As seen though in Table 3, but not really touched on in the discussion, there is a moderately important comment from the stakeholders that the 11 faces look too similar and that the frequently used 6 face scale should be used. Yes, exactly, this is most likely referring to the Faces Pain Scale - Revised. Now, I understand you are using a new 11-faces pain scale to match the 11 numerical values. Amazingly, the paper that you cite for this, Van Giang et. al 2015 in Pain Management Nursing does not have an image of the scale. The original 11-face FPS paper, cited in that paper as Kim and Buschmann, 2016 in International Journal of Nursing Studies has 11 faces that appear to have more detail and look different than the very simple faces you used. Where did your faces come from? Were they designed by you for this app? That is not implied by your text. It would be best to use existing scales in the app to rely on some existing scientific evidence. Additionally, there is lots of good and critical feedback in Table 3 that can be used to improve your app. Perhaps once that is done, you can do a better test with it.

2. Leading from the above point, the second major problem is that the secondary objective is completely mixed up with the primary objective. That is, comparing the pain scores from this app with the nurse scores is not really meaningful, if you are now going to make large adjustments on the app - thus one of the two things you are comparing is disappearing (being greatly altered) in the future. If you are going to change the pain scale, which you should, then the comparison is not meaningful. I would suggest removing this secondary objective.

I also found the secondary objective results confusing, as you say that the patient and nurse scores are similar enough that one could substitute for the other (based on no significant difference in the medians) but then dive into the details of how certain ranges on the scales are significantly different.

Minor issues:

I'm not sure how relevant these below points will be given the major issues, but I have included them anyway. I understand that it can be quite an exercise to attempt to publish this work in English when it was all done in Dutch, but please take more care in the future to have the manuscript carefully looked through internally before you submit.

3. Please double check your numbers for the results. You have an error for the same number in the abstract and the manuscript body, assuming your table is correct. In the abstract

"‘severe’: 24 (28%)." should be "‘severe’: 14 (28%)."

In the body:

"and 30 (28%) valued their pain as considerable to severe" should be "and 14 (28%) valued their pain as considerable to severe"

4. Please be clearer about where (and how many places) the study took place. You currently have:

"study was conducted in a Dutch general hospital situated at two locations in Amsterdam, the Netherlands"

Ok...so this is two locations of the same hospital? Or is this two areas of one hospital? In the discussion (limitations paragraph) you have:

"the study was performed in a single center setting"

5. Some typos/grammar problems in the manuscript:

Participants and recruitment:

"...provided them instructions how to use the application..." should be:

"...provided them instructions on how to use the application..."

"...A patient sample size of 50 patients was, based on sample size calculation for qualitative studies, was estimated..." should be:

"...A patient sample size of 50 patients, based on sample size calculation for qualitative studies, was estimated..."

Figure 4: "adndroid" and "particoipate".

Discussion:

"under the implementation of patients self-recording pain on a lager scale."

should be:

"under the implementation of patients self-recording pain on a larger scale."

"If a patient answer is yes, a comment for the nurses to discus the pain appears in the patient electronic medical record."

Should be:

"If a patient answer is yes, a comment for the nurses to discuss the pain appears in the patient electronic medical record."

Conclusion:

"...high satisfaction rate for the majority of patients stakeholders and that it provides outcome comparable to nurses pain assessment"

Should be:

"...high satisfaction rate for the majority of patients stakeholders and that it provides an outcome comparable to nurses pain assessment."

6. Patients were notified three times daily to assess their pain. At what times? Was it different per patient based on when they started or at the same time for all patients?

7. This phrasing is confusing:

"We determined the following items from the literature: design, usability, content, and workflow indexing the feedback from the patients and stakeholders"

I think you mean to say that you grouped feedback into these themes or categories, as has been done before (by cited papers).

8. How come the percentages in the rows of Table 2 all add up to different amounts (below 100)? Do these represent patients who did not reply to these questions? You have the brackets wrong in the cell that's in the 10th row and 3rd column.

9. Part of the Discussion reads:

"the number of pain scores ranging from NRS ‘0 to 4’ and NRS ‘5 to 7’ recorded by patients were statistically significant higher compared with the recordings of nurses (p 0.0024, p 0.0096)"

This is not really written correctly. You should say "the percentage" not "the number" and the '0 to 4' range was lower while the '5 to 7' range was higher than when recorded by nurses, not both higher.

10. Reviewer #2 has a good point, that "The results need to provide descriptive statistics-including minimum and maximum – for the number of ratings provided per patient before questionnaire completion". I do not understand your response. The reviewer is referring to statistics summarizing the number of ratings per participant using the app. By "questionnaire" in your response do you mean the app ratings or the end questionnaire? Either you are saying that the data in the app does not have time stamp (which can't be true since you provide an in-app graph of ratings over time) or perhaps you are saying the questionnaire was completed before the actual end use of the app...ie. patients used the app even after the questionnaire. This second possibility would be quite odd, but regardless you could summarize the total number of ratings done by each patient.

11. Later Reviewer #2 also asks "For example, were they interviewed individually or in groups?". You should put the answer to this question into your actual manuscript as it is an important point.

12. Finally, I see that you said "We would also like to mention that the tables and reference list are only correct in the clean version copy, as we made changes to it which could not adequately be processed in the track changes version."

Unfortunately, in the regular body of the manuscript (not in tables and references) there are a number of difference between the tracked version and the cleaned version. It seems some things were adjusted in both separately. This makes it difficult to review. Some examples are on lines 170-171, 315, 352-353.

7. PLOS authors have the option to publish the peer review history of their article (what does this mean?). If published, this will include your full peer review and any attached files.

Reviewer #3: No

---

## [Author Response · Author response to Decision Letter 1]

3 Feb 2020

Rebuttal letter: PONE-D-19-17106R1

Title: Patient reported postoperative pain with a smartphone application: a proof of concept

Date: January 31st 2020

Dear Editorial Board, Dear Peter ten Klooster,

On behalf of all authors who contributed to this manuscript I thank you for the opportunity to submit a revised version. We are grateful to the editor and the reviewer for their valuable comments and suggestions for improvement. Below we address each comment separately. 

Editorial comments:

Please pay particular attention to and thoroughly check the accuracy of all statistics throughout the entire manuscript. The reviewer already pointed out several errors or inconsistencies between the abstract and the manuscript body. I also noted a potential error with the numbers in the abstract. For instance, in the abstract you described that “…Two patients (1%) were …”, whereas this percentage should probably be 4% considering the sample of 50 patients. 

The editor and the reviewer identified several inconsistencies in the descriptive statistics. We performed a thorough check of all descriptives in the abstract and the body of the manuscript. We found inconsistencies in the abstract, the body of the manuscript and table 2 and corrected these. From the abstract we have removed the following ‘Experienced pain after surgery was scored by patients as ‘no to mild’: 8 (16%), ‘bearable’: 25 (50%) and ‘considerable’ or ‘severe’: 24 (28%) (track changes lines 36-37). 

In the body of the manuscript we corrected the following in line with table 2 ‘The overall experienced postoperative pain was valued as no pain by 3 patients (6%), or little pain in 5 patients (10%), 25 patients (50%) valued their pain as bearable and 13 (26%) valued their pain as considerable. One patient (2%) experienced severe pain’ (track changes lines 239-242). 

In table 2 we have corrected the cell in row 12, column 3 from: 24 (48%) in 23 (46%). Thereby we also corrected this sentence from the results section: ‘The in-app pain intensity chart was valued as (very) useful by 41 patients (82%) in ‘The in-app pain intensity chart was valued as (very) useful by 40 patients (80%)’ (track changes lines 247-248).

To enhance the reproducibility of your results, we recommend that if applicable you deposit your laboratory protocols in protocols.io 

As already mentioned in the previous rebuttal letter a summary of the study protocol is published in the Dutch trial register (www.trialregister.nl). The Dutch trial register is a publicly accessible and freely searchable register for prospective studies run in the Netherlands or that are carried out by Dutch researchers. The study registration number is: NL6565 (track changes lines 91-93). 

Reviewer comments:

1.The revisions to the manuscript have added the label of NRS or Numerical Rating Scale to describe the app pain scale used. This is just not accurate. The Numerical Rating Scale is not just using the numbers 0-10 to describe pain. It should include these numbers along a line, with the whole line visible at once. Often there are anchor words at either end. The app, as shown in Figure 1, does not do this. I assume a line with number is what is used by the nurse. If instead the nurse used some other form of the numbers or it was purely verbal, it would be good to add this to the manuscript. Your Kim et al 2012 citation from the Journal of Palliative Care clearly shows a line with numbers scale where you can see the whole line at once. Your first citation, for the NRS, Breivik et al. 2008 from BJA also shows a line with number for the NRS. 

Please refrain from referring to the app scale as the NRS. Much work has been done to validate the proper NRS as a good pain scale. Is there any scientific evidence of this wheel scale (where you can only see a few numbers at once and the faces seem to be the focus) being consistent and valid as a pain scale? What was the starting position of the wheel? Was it 0, 5, or 10? This needs to be well explained or cited within the paper. 

We agree with the reviewer that referring to the pain app scale as numerical rating scale (NRS) is not appropriate. Indeed Breivik et al. 2008 shows a line with numbers for the NRS but they also mention in their paper ‘An NRS with numbers from 0 to 10 (‘no pain’ to ‘worst pain imaginable’) is more practical than a VAS, easier to understand for most people, and does not need clear vision, dexterity, paper, and pen. One can even determine the intensity of pain accurately using telephone interview, a computerized telephone interview, and recording of NRS data by the patient directly into the database of a computer via the telephone keyboard’ this is why we chose to refer to the pain app scale as NRS. In the manuscript we now use the neutral term ‘pain app scale’ and have corrected this throughout the manuscript. 

The nurses in OLVG hospital routinely assess postoperative pain verbally by asking patients to indicate the severity of their pain with a number from 0 to 10, which is common practice in most Dutch hospitals. This is now mentioned in the manuscript : ‘Moreover, postoperative pain intensity was regularly verbally assessed on a scale from 0 (no pain) to 10 (worst imaginable pain) by ward nurses at least once every eight hours and thereafter recorded in the patient’s electronic medical record (EMR). This method of pain assessment is common practice in most Dutch hospitals and commonly referred to as ‘numerical rating scale’ (track changes lines 127-132). The starting position of the wheel was default set at 5, this in now mentioned in the manuscript: ‘The default starting position of the wheel was set at 5 and the wheel was provided with text anchors at both ends of the scale referring to no pain (0) and worst imaginable pain (10)’ (track changes lines 145-147).

As seen though in Table 3, but not really touched on in the discussion, there is a moderately important comment from the stakeholders that the 11 faces look too similar and that the frequently used 6 face scale should be used. Yes, exactly, this is most likely referring to the Faces Pain Scale - Revised. Now, I understand you are using a new 11-faces pain scale to match the 11 numerical values. Amazingly, the paper that you cite for this, Van Giang et. al 2015 in Pain Management Nursing does not have an image of the scale. The original 11-face FPS paper, cited in that paper as Kim and Buschmann, 2016 in International Journal of Nursing Studies has 11 faces that appear to have more detail and look different than the very simple faces you used. Where did your faces come from? Were they designed by you for this app? That is not implied by your text. It would be best to use existing scales in the app to rely on some existing scientific evidence. Additionally, there is lots of good and critical feedback in Table 3 that can be used to improve your app. Perhaps once that is done, you can do a better test with it.

We agree with the comments from the reviewer on both the pain scale and faces scale used the smartphone application. Both the pain scale and 11-faces scale were designed for this application. The faces scale used in the application was designed to fit with the 11 point numerical scale and was based on the 6 faces scale recommended by the 2015 pain guideline from the Dutch college of General Practitioners. This is now mentioned in the manuscript with reference (track changes lines 149-151). We disagree that it is not useful to test with both the pain scale and faces scale in this phase of developing an application. It is very important early in the design of eHealth tools to use end-user feedback to further improve the application one of the objectives of the study. 

We agree that the used pain scale and 11-faces scale are not validated yet. We now mention this in the abstract and manuscript body ’Further research is needed to validate in-app pain scale and the 11-faces scale with the current gold standard VAS and NRS for pain’ (track changes lines 48-49) ‘More importantly, the pain intensity wheel has not yet been validated in relation to VAS or NRS, which are currently considered the gold standard for pain assessment (track changes lines 356-358) ‘Much work still needs to be done to thoroughly adjust and examine the psychometric features of the in-app pain intensity wheel and validate it against VAS and NRS’ (track changes lines 404-406).

2. Leading from the above point, the second major problem is that the secondary objective is completely mixed up with the primary objective. That is, comparing the pain scores from this app with the nurse scores is not really meaningful, if you are now going to make large adjustments on the app - thus one of the two things you are comparing is disappearing (being greatly altered) in the future. If you are going to change the pain scale, which you should, then the comparison is not meaningful. I would suggest removing this secondary objective. 

We understand the reviewer on this point but we do not fully agree and we believe we have valid arguments to underpin this; the objective of this study is to collect data for improvement of the application and to see if patient self-recording of postoperative pain with a smartphone application could be used for postoperative pain management. We believe it is appropriate, even necessary, in this early stage of developing a smartphone application, to assess the agreement between patient-recorded and nurse-recorded pain scores, to better understand what is needed to improve and validate the used pain scale and faces scale. 

I also found the secondary objective results confusing, as you say that the patient and nurse scores are similar enough that one could substitute for the other (based on no significant difference in the medians) but then dive into the details of how certain ranges on the scales are significantly different.

We agree with the reviewer that the results of the secondary objectives might confuse the reader. Therefore we have decided to remove the following paragraphs (including figure 5) from the manuscript to provide more clarity: 

‘Patients record their pain more widely distributed throughout day and night (Figure 5). Nurses record pain mainly during their daily rounds, at 9:00, 12:00, and 18:00 and between 21:00 and 22:00. During the night, from 0:00 to 3:00, patients recorded pain 20 times and nurses recorded pain 4 times, respectively with median NRS 5 (range 1 to 9) and median NRS 6.5 (range 3 to 7) with a difference of 1.5 (p = 0.723). During the night, from 0:00 to 8:00, patients recorded their pain 54 times and nurses recorded pain 57 times, respectively with median NRS 5 (0 to 9) and median NRS 3 (0 to 8) with a difference of 2 (p = 0,03)’. Figure 5: Time distribution of pain score entries (track changes lines 303-310).

‘More importantly, the pain scores recorded by patients during the night were statistically significant higher compared with those recorded by nurses. Those patients with nightly high pain scores could potentially benefit from using the pain application, especially if we are able to establish a feedback loop from the app to the patients’ electronic medical record combined with an automated on-demand nurse warning system. However, in a recent study amongst 105 chronic pain patients, pilot testing a smartphone application with and without 2-way messaging between patient and healthcare provider. Patients in the 2-way massaging group felt that their healthcare providers were less responsive (29). Therefore it seems very important that the nurse is able to respond to the patients call when introducing 2-way messaging or a feedback loop in daily hospital care’ (track changes lines 342-351).

Minor issues:

3. Please double check your numbers for the results. You have an error for the same number in the abstract and the manuscript body, assuming your table is correct. In the abstract

"‘severe’: 24 (28%)." should be "‘severe’: 14 (28%)."

In the body:

"and 30 (28%) valued their pain as considerable to severe" should be "and 14 (28%) valued their pain as considerable to severe"

Thank you for pointing this out. We have corrected the descriptive statistics in the abstract (track changes lines 29-30) and the manuscript body (track changes lines 239-242), (track changes lines 247-248), according to the results from table 2.

4. Please be clearer about where (and how many places) the study took place. You currently have:

"study was conducted in a Dutch general hospital situated at two locations in Amsterdam, the Netherlands"

Ok...so this is two locations of the same hospital? Or is this two areas of one hospital? In the discussion (limitations paragraph) you have:

"the study was performed in a single center setting"

The study was conducted in OLVG Hospital which has two locations situated in different parts of Amsterdam. We adjusted the discussion section with the following sentences ‘This study is not without limitations. First, the study was performed on two locations of OLVG Hospital situated at different parts in Amsterdam which may not benefit the generalizability of the results’ (track changes lines 377-379).

5. Some typos/grammar problems in the manuscript:

"...provided them instructions how to use the application..." should be:

"...provided them instructions on how to use the application..."

This has been corrected (track changes line 108).

"...A patient sample size of 50 patients was, based on sample size calculation for qualitative studies, was estimated..." should be:

"...A patient sample size of 50 patients, based on sample size calculation for qualitative studies, was estimated..."

This has been corrected (track changes line 109).

Figure 4: "adndroid" and "particoipate".

This has been corrected, together with other adjustments of the figure.

"under the implementation of patients self-recording pain on a lager scale." 

should be:

"under the implementation of patients self-recording pain on a larger scale."

This has been corrected (track changes line 401).

"If a patient answer is yes, a comment for the nurses to discus the pain appears in the patient electronic medical record."

Should be:

"If a patient answer is yes, a comment for the nurses to discuss the pain appears in the patient electronic medical record."

This has been corrected (track changes line 409).

"...high satisfaction rate for the majority of patients stakeholders and that it provides outcome comparable to nurses pain assessment"

Should be:

"...high satisfaction rate for the majority of patients stakeholders and that it provides an outcome comparable to nurses pain assessment." 

This is corrected (track changes line 419).

6. Patients were notified three times daily to assess their pain. At what times? Was it different per patient based on when they started or at the same time for all patients? 

All patients were notified at fixed times at 8:00 am, 14:00 pm and 22:00 pm. We adjusted the text into ‘In addition, all patients were notified three times daily fixed at 8:00 am, 14:00 pm and 22:00 pm by the application to record pain’ (track changes lines 139-141).

7. This phrasing is confusing: "We determined the following items from the literature: design, usability, content, and workflow indexing the feedback from the patients and stakeholders"

I think you mean to say that you grouped feedback into these themes or categories, as has been done before (by cited papers). 

We adjusted the phrasing into ‘we grouped feedback from patients and stakeholders into the following themes determined the following items from the literature: design, usability, content, and workflow, obtained from previous research indexing the feedback from the patients and stakeholders (25-26)’ track changes lines 186-188).

8. How come the percentages in the rows of Table 2 all add up to different amounts (below 100)? Do these represent patients who did not reply to these questions? You have the brackets wrong in the cell that's in the 10th row and 3rd column.

Indeed, the different percentages in the rows of table 2 are due to the fact that 3 patients did not or partially complete the questionnaire. This was already mentioned in the text (track changes lines 207-210). We have now added: ‘one patient used the application, but only partially completed the questionnaire’ (track changes lines 210-211). Furthermore, we adjusted the table 2 legend with the following sentence ‘the difference in total numbers per question is explained by 3 patients who did not or partially complete the questionnaire’ (track changes line 250-251). The brackets in cell row 10, column 3 have been corrected 

9. Part of the Discussion reads: "the number of pain scores ranging from NRS ‘0 to 4’ and NRS ‘5 to 7’ recorded by patients were statistically significant higher compared with the recordings of nurses (p 0.0024, p 0.0096)"

This is not really written correctly. You should say "the percentage" not "the number" and the '0 to 4' range was lower while the '5 to 7' range was higher than when recorded by nurses, not both higher.

Thank you. We have corrected the text to: ‘However, there are some differences, the percentage of pain scores ranging from ‘0 to 4’ recorded by patients with the app was lower compared with the NRS recordings of the nurses (p = 0.0024). Moreover, the percentage of pain scores ranging from ‘5 to 7’ recorded by patients with the app was higher compared with the NRS recordings of nurses (p = 0.0096) although these differences are probably not clinically relevant’ (track changes lines 334-340).

10. Reviewer #2 has a good point, that "The results need to provide descriptive statistics-including minimum and maximum – for the number of ratings provided per patient before questionnaire completion". I do not understand your response. The reviewer is referring to statistics summarizing the number of ratings per participant using the app. By "questionnaire" in your response do you mean the app ratings or the end questionnaire? Either you are saying that the data in the app does not have time stamp (which can't be true since you provide an in-app graph of ratings over time) or perhaps you are saying the questionnaire was completed before the actual end use of the app...ie. patients used the app even after the questionnaire. This second possibility would be quite odd, but regardless you could summarize the total number of ratings done by each patient.

We agree on this point that our answer was not clear. In this current revision we have now provided descriptive statistics for the number of ratings per patient before questionnaire completion in the Results section: ‘Indicating the experience patients have gained in using the app. The median number of pain recordings by patients before questionnaire completion was 3 with a range of 0 to 13’ (track changes lines 214-216).

11. Later Reviewer #2 also asks "For example, were they interviewed individually or in groups?” You should put the answer to this question into your actual manuscript as it is an important point.

We agree with the reviewer that this is important information. It was already mentioned in the manuscript. ‘Stakeholders were individually questioned during a semi-structured interview conducted by a researcher after they had the opportunity to examine the application (supporting information 3 and 4)’ (track changes lines 173-175).

12. Unfortunately, in the regular body of the manuscript (not in tables and references) there are a number of differences between the tracked version and the cleaned version. It seems some things were adjusted in both separately. This makes it difficult to review. Some examples are on lines 170-171, 315, 352-353. 

We want apologize for making the reviewing process more complicated. We have now corrected all identified errors and carefully addressed all editorial and reviewer comments in the current track changes version of the manuscript and made an exact copy of the changes in the ‘clean version’.

---

## [Decision Letter · Decision Letter 2]

18 Feb 2020

PONE-D-19-17106R2

Patient reported postoperative pain with a smartphone application: a proof of concept

PLOS ONE

Dear Thiel,

Thank you for submitting your manuscript to PLOS ONE. After careful consideration, we feel that it has merit but does not fully meet PLOS ONE’s publication criteria as it currently stands. Therefore, we invite you to submit a revised version of the manuscript that addresses the points raised during the review process.

Your manuscript was re-reviewed by the same reviewer. As you can see from the comments below, the reviewer was generally satisfied with the revisions made in response to the previous major issues. Some minor textual issues remain, which should be relatively easy to adjust.

We would appreciate receiving your revised manuscript by Apr 03 2020 11:59PM. To enhance the reproducibility of your results, we recommend that if applicable you deposit your laboratory protocols in protocols.io, where a protocol can be assigned its own identifier (DOI) such that it can be cited independently in the future. For instructions see: http://journals.plos.org/plosone/s/submission-guidelines#loc-laboratory-protocols

We look forward to receiving your revised manuscript.

Kind regards,

Peter M ten Klooster, Ph.D.

Academic Editor

PLOS ONE

Reviewers' comments:

Reviewer's Responses to Questions

**Comments to the Author**

1. If the authors have adequately addressed your comments raised in a previous round of review and you feel that this manuscript is now acceptable for publication, you may indicate that here to bypass the “Comments to the Author” section, enter your conflict of interest statement in the “Confidential to Editor” section, and submit your "Accept" recommendation.

Reviewer #3: (No Response)

2. Is the manuscript technically sound, and do the data support the conclusions?

Reviewer #3: Yes

3. Has the statistical analysis been performed appropriately and rigorously? 

Reviewer #3: Yes

4. Have the authors made all data underlying the findings in their manuscript fully available?

Reviewer #3: Yes

5. Is the manuscript presented in an intelligible fashion and written in standard English?

Reviewer #3: Yes

6. Review Comments to the Author

Reviewer #3: Thank you to the authors for addressing my comments. Figure 4 has been much improved. The removal of some of the secondary outcome text strengths the paper. There are only minor revisions to be made now.

Regarding the response to my previous point about the primary and secondary objectives being entangled, I agree with the authors that this study has clearly helped them understand what is needed to improve the app and the pain and faces scale, although I fail to see how comparing the resulting scores to nurse-recorded pain scores contributes to this. The goal of that original secondary objective was clearly to show the benefit of the app, rather than find its weaknesses. Nonetheless this can stay mentioned in the paper as I find the removal of the confusing parts of the secondary outcome adequate. Removing the paragraphs and Figure 5 narrows the focus of the paper, which is an improvement.

Since the results paragraph about when pain scores were done (during the night, etc.) has been removed, please remove the 2 sentences from the discussion, lines 340-342 as they do not make sense given the lack of associated data in the shortened results.

You have not actually removed all mentions of your in-app scale as an NRS from your paper. This should still be corrected in the Secondary outcome section, where for example on line 292 of the Tracked Changes version you refer to the "patient recorded NRS". Please adjust this section accordingly.

Additionally, the authors refer to "the in-app pain scale and 11-faces scale" in the abstract, but from what I see in Figure 1 the numbers and faces all go together on a single wheel (are moved as one), so the whole thing should be referred to as the "in-app pain scale" or "in-app pain wheel". At the end of the abstract, it could be referred to as the "in-app pain scale with 11 faces" or "the app's 11 point numeric and faces pain scale" or either option with "wheel" instead of "scale".

In the Study Procedures section, thank you for adding the notification times on lines 139 and 140 as this fits with the three dots per day in Figure 3. Please remove the "am" and "pm" from these lines, as you are using a 24-hour clock, so it is unnecessary.

In the Results and Discussion - Main Outcome section, I take issue with the way that you have labelled the lumped together responses for "satisfying and very satisfying" as well as "agree and totally agree" by using brackets around "very" and "totally". It is not immediately obvious that you are giving the sum of these two responses. I would much prefer you change "(very) satisfying" to "satisfying or very satisfying" and "(totally) agreed" to "agreed or totally agreed". This is how you listed it in the abstract.

Please fix your new reference 19, which is missing "http" in the URL.

Some small grammar issues to adjust:

On line 30 of the tracked changes version, thank you for fixing the counts and percents, but you have two "and"s in the sentence. The first one should just be a comma.

Line 131, in order to make this sentence grammatical correct please change it to "referred to as a 'numerical rating scale'" or "referred to as the 'numerical rating scale'". If using "the", perhaps you should capitalize it: 'Numerical Rating Scale'.

Line 214 onto 215, you have an odd sentence fragment: "Indicating the experience patients have gained in using the app." This is not a complete sentence because it does not have a subject. Please combine this with the next sentence in some way so that it is grammatically correct.

Please add a closing single quote on line 332 to end the quote from the patient.

In the discussion, your points are numbered, but there is no "Four". Please fix this by changing "Five," to "Four," on line 371.

Thanks for the clarification on line 377, but please change "on two locations" to "in two locations" or "at two locations" as these are proper ways to say it in English. Also please change "situated at different parts in Amsterdam" to "situated at different parts of Amsterdam" or simply "in Amsterdam".

Remove the extra period on line 388 and add a period at the end of your conclusion.

7. PLOS authors have the option to publish the peer review history of their article (what does this mean?). If published, this will include your full peer review and any attached files.

Reviewer #3: No

---

## [Author Response · Author response to Decision Letter 2]

6 Apr 2020

Editorial comments:

To enhance the reproducibility of your results, we recommend that if applicable you deposit your laboratory protocols in protocols.io 

As already mentioned in the previous rebuttal letter a summary of the study protocol is published in the Dutch trial register (www.trialregister.nl). The Dutch trial register is a publicly accessible and freely searchable register for prospective studies run in the Netherlands or that are carried out by Dutch researchers. The study registration number is: NL6565. (Track changes lines 85-87)

Reviewers' comments:

1. Since the results paragraph about when pain scores were done (during the night, etc.) has been removed, please remove the 2 sentences from the discussion, lines 340-342 as they do not make sense given the lack of associated data in the shortened results.

Removed: Moreover, patients recorded their pain more divided through the day and more during the night compared to nurses. Nurses tend to record pain during specific moments as part of their daily routine and care rounds for patients. (Tack changes lines 331-333)

2. You have not actually removed all mentions of your in-app scale as an NRS from your paper. This should still be corrected in the Secondary outcome section, where for example on line 292 of the Tracked Changes version you refer to the "patient recorded NRS". Please adjust this section accordingly.

Adjusted into: Patients recorded 307 times their pain score with the app, while nurses recorded 396 times a NRS for pain. The median patient recorded pain app score was 4.0 (range 0 to 10), the median nurse recorded NRS for pain was 4.0 (range 0 to 9) p = 0.06. The differences in pain recordings ranging from ‘0 to 4’, ‘5 to 7’ and ‘8 to 10’ between patients and nurses were respectively 11% (p = 0.0024), 9% (p = 0.0096) and 2% (p = 0.27) (Table 4). One hundred ninety seven patient pain app scores (64%) were rated as pain during rest. Patients asked 92 times (29%) for extra analgesia and 63 times (20%) for a nurse. One hundred thirty one patient pain app scores (42%) were ≥ 5, of these, 109 were still rated as bearable and only 55 times they asked for extra analgesia (Table 5). (Track changes lines 289-297)

3. Additionally, the authors refer to "the in-app pain scale and 11-faces scale" in the abstract, but from what I see in Figure 1 the numbers and faces all go together on a single wheel (are moved as one), so the whole thing should be referred to as the "in-app pain scale" or "in-app pain wheel". At the end of the abstract, it could be referred to as the "in-app pain scale with 11 faces" or "the app's 11 point numeric and faces pain scale" or either option with "wheel" instead of "scale".

Adjusted the following sentence in abstract: Further research is needed to validate the 11-point numeric and faces pain scale in-app pain scale and the 11-faces scale with the current gold standards visual analogue scale (VAS) and NRS for pain. (Track changes lines 45-46)

Adjusted sentence in the discussion: Much work still needs to be done to thoroughly adjust and examine the psychometric features of the app’s 11-point numeric and faces pain scale and validate it against VAS and NRS. (Track changes lines 386-387)

4. In the Study Procedures section, thank you for adding the notification times on lines 139 and 140 as this fits with the three dots per day in Figure 3. Please remove the "am" and "pm" from these lines, as you are using a 24-hour clock, so it is unnecessary.

Removed: “am” and “pm” (Track changes lines 132-133)

5. In the Results and Discussion - Main Outcome section, I take issue with the way that you have labelled the lumped together responses for "satisfying and very satisfying" as well as "agree and totally agree" by using brackets around "very" and "totally". It is not immediately obvious that you are giving the sum of these two responses. I would much prefer you change "(very) satisfying" to "satisfying or very satisfying" and "(totally) agreed" to "agreed or totally agreed". This is how you listed it in the abstract.

We have changed it into the following: Thirty patients (60%) rated communicating the degree of pain with the application as satisfying or very satisfying (Table 2). The overall experienced postoperative pain was valued as no pain by 3 patients (6%), little pain in 5 patients (10%), 25 patients (50%) valued their pain as bearable and 13 (26%) valued their pain as considerable. One patient (2%) experienced severe pain. Asking patients if they could easily and correctly record their pain with the application 45 (90%) agreed or totally agreed. Asking about the three times daily notifications to score pain 38 patients (76%) agreed or totally agreed that this was useful. Regarding the overall appearance of the app 40 patients (80%) found it attractive or very attractive. Asking if it would be beneficial to contact a nurse with the application 9 (18%) of the patients reported that it would not be beneficial. The in-app pain intensity chart was valued as useful or very useful by 40 patients (80%). (Track changes lines 232-242)

6. Please fix your new reference 19, which is missing "http" in the URL.

Fixed reference 18 

18. Richtlijn Postoperatieve Pijn 2012 [Available from: http://www.anesthesiologie.nl].

Fixed reference 19

19. De NHG-Standaard PIJN 2015 [Available from: http://www.nhg.org/standaarden/volledig/nhg-standaard-pijn].

(Track changes lines 436 and 437)

7. On line 30 of the tracked changes version, thank you for fixing the counts and percents, but you have two "and"s in the sentence. The first one should just be a comma.

Corrected into: Pain experienced after surgery was scored by patients as ‘no’: 3 (6%), ‘little’: 5 (10%), ‘bearable’: 25 (50%), ‘considerable’: 13 (26%) and ‘severe’: 1 (2%). (Track changes lines 29-30)

8. Line 131, in order to make this sentence grammatical correct please change it to "referred to as a 'numerical rating scale'" or "referred to as the 'numerical rating scale'". If using "the", perhaps you should capitalize it: 'Numerical Rating Scale'.

Corrected into: This method of pain assessment is common practice in most Dutch hospitals and commonly referred to as a ‘numerical rating scale’. (Track changes lines 124-125)

9. Line 214 onto 215, you have an odd sentence fragment: "Indicating the experience patients have gained in using the app." This is not a complete sentence because it does not have a subject. Please combine this with the next sentence in some way so that it is grammatically correct.

Corrected into: Indicating the experience patients have gained in using the app, the median number of pain app recordings before questionnaire completion was 3 (range 0 to 13). (Track changes lines 209-211)

10. Please add a closing single quote on line 332 to end the quote from the patient.

Added: One patient stated that ‘When I need a nurse immediately I’ll use the button next to my bed’. (Track changes line 323)

Added: He said: ‘I only watch the red notifications on the corners of the app icon’. (Track changes line 342) 

11. In the discussion, your points are numbered, but there is no "Four". Please fix this by changing "Five," to "Four," on line 371.

Corrected into: Four, to add more and clear personalized feedback messages in the app. (track changes line 353)

12. Thanks for the clarification on line 377, but please change "on two locations" to "in two locations" or "at two locations" as these are proper ways to say it in English. Also please change "situated at different parts in Amsterdam" to "situated at different parts of Amsterdam" or simply "in Amsterdam".

Corrected into: First, the study was performed at two locations of OLVG Hospital in Amsterdam…(Track changes lines 359-360)

13. Remove the extra period on line 388 and add a period at the end of your conclusion.

Removed and added a period (track changes lines 369 and 401)

---

## [Editor Report · Decision Letter 3]

8 Apr 2020

Patient reported postoperative pain with a smartphone application: a proof of concept

PONE-D-19-17106R3

Dear Dr. Thiel,

We are pleased to inform you that your manuscript has been judged scientifically suitable for publication and will be formally accepted for publication once it complies with all outstanding technical requirements.

With kind regards,

Peter M ten Klooster, Ph.D.

Academic Editor

PLOS ONE

---

## [Editor Report · Acceptance letter]

27 Apr 2020

PONE-D-19-17106R3 

Patient reported postoperative pain with a smartphone application: a proof of concept 

Dear Dr. Thiel:

I am pleased to inform you that your manuscript has been deemed suitable for publication in PLOS ONE. Congratulations! Your manuscript is now with our production department. 

With kind regards,

on behalf of

Dr. Peter M ten Klooster 

Academic Editor

PLOS ONE